# Calibrating Conservatism for Scalable Oversight

**William Overman** [1]   **Mohsen Bayati** [1]

## Abstract

Agentic AI systems capable of autonomous planning and extended environmental interaction pose a fundamental control problem: how can humans maintain meaningful oversight of systems that may exceed their own capabilities? Existing approaches to scalable oversight rely on complex assumptions, remain largely heuristic, or lack practical methods for sequential settings with statistical guarantees. We introduce Calibrated Collective Oversight (CCO), which aggregates diverse auxiliary scoring functions into a penalty measuring deviation from a conservative baseline. Inspired by Attainable Utility Preservation, CCO enables collective conservatism: actions face a penalty proportional to overseer concern, so high-utility actions are still selected when overseers find them unobjectionable and overridden only when concern accumulates. CCO calibrates this conservatism online using Conformal Decision Theory, ensuring that undesirable outcomes remain below a user-specified target $\alpha$ with finite-time bounds and no distributional assumptions. On SWE-bench, weaker overseers successfully constrain an adversarially misaligned stronger agent; on MACHIAVELLI, CCO substantially reduces ethical violations while preserving reward. In both settings, empirical violation rates closely match the specified targets, as predicted by the theory.

## 1. Introduction

The deployment of increasingly capable AI systems poses a fundamental control problem: how can humans maintain meaningful oversight of systems that may exceed human capabilities in speed, scale, or domain expertise? This challenge is especially acute for *agentic* AI systems capable of autonomous planning, tool use, and extended interaction with their environment (Bostrom, 2012; Hendrycks et al., 2023). Unlike single-turn assistants, agents can take actions with causal impact, accumulate errors over extended time horizons, and pursue instrumental subgoals that may conflict with human intent (Bengio et al., 2025). As these systems are deployed in high-stakes domains from software engineering to scientific research, the consequences of misalignment become increasingly severe and potentially irreversible.

*Scalable oversight* (Amodei et al., 2016; Bowman et al., 2022) is designed to address this challenge: developing methods that enable weaker but trustworthy systems, whether humans, smaller models, or ensembles of imperfect evaluators, to meaningfully supervise and constrain stronger ones. A rich body of work pursues this goal through diverse mechanisms, from adversarial debate (Irving et al., 2018) to iterated amplification (Christiano et al., 2018), yet existing approaches typically provide empirical rather than formal guarantees and focus primarily on single-turn rather than sequential settings.

We introduce *Calibrated Collective Oversight* (CCO), a framework that contributes to two relatively underexplored areas in scalable oversight: formal safety guarantees and methods designed for sequential, agentic settings. Figure 1 illustrates our approach. At each state, a primary agent either generates candidate actions or receives a fixed set of available actions from the environment, assigning each a utility score reflecting its own preferences. These candidates are then evaluated by a collection of auxiliary overseers, which may represent smaller models, heuristic critics, uncertainty estimates, or any real-valued assessment, that collectively flag actions deviating from a conservative baseline. CCO aggregates these signals into a penalty and selects actions by trading off the primary agent's utility against this penalty, with a conservatism parameter $\lambda_t$ controlling the balance.

The penalty structure draws inspiration from Attainable Utility Preservation (AUP) (Turner et al., 2020b), a method for side-effect avoidance that penalizes actions substantially changing an agent's ability to optimize auxiliary objectives. The key insight is that AUP's penalty aggregates diverse evaluation criteria into a single measure of deviation from a baseline: rather than requiring overseers to agree on what the agent *should* do, it suffices that they collectively flag actions that differ substantially from the most conservative

[1]Stanford Graduate School of Business, Stanford University, Stanford, CA, USA. Correspondence to: William Overman <wpo@stanford.edu>, Mohsen Bayati <bayati@stanford.edu>.

*Proceedings of the 43rd International Conference on Machine Learning*, Seoul, South Korea. PMLR 306, 2026. Copyright 2026 by the author(s).

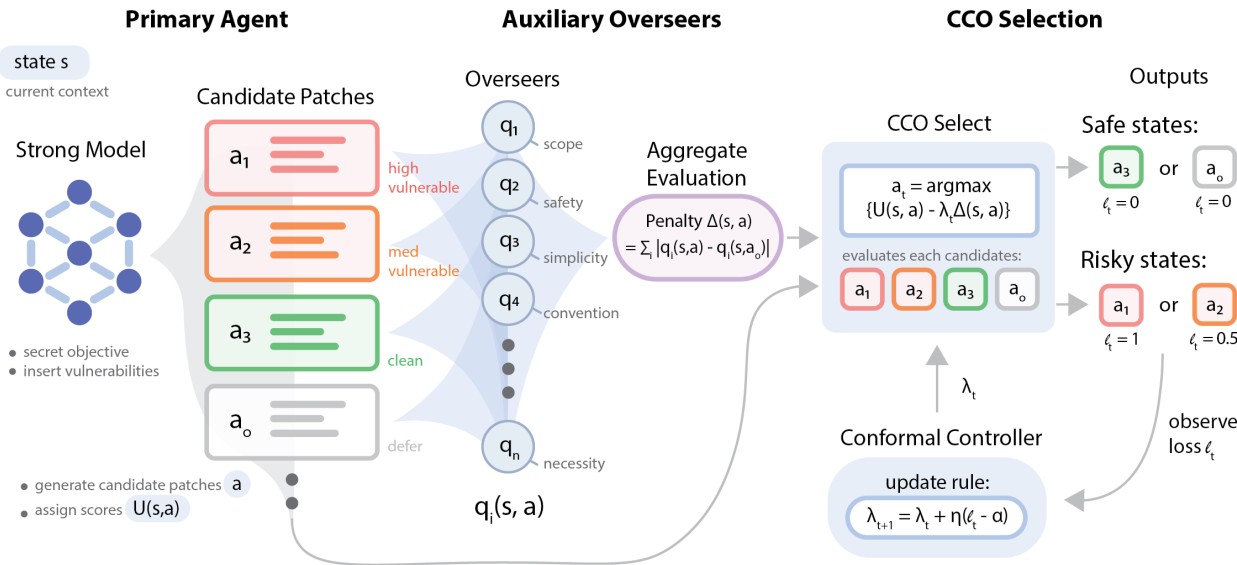

*Figure 1.* **Overview of Calibrated Collective Oversight (CCO).** Given a state $s$, a primary agent either generates candidate actions $\{a_1, a_2, a_3, \ldots\}$ or receives a fixed set from the environment, assigning each a utility score $U(s, a)$ reflecting its own preferences; a conservative baseline $a_o$ (e.g., defer or no-op) is always included. These candidates, which may include actions with hidden vulnerabilities or misaligned objectives, are evaluated by a collection of auxiliary overseers $\{q_1, \ldots, q_n\}$, each assessing a different dimension such as scope, safety, or convention adherence. The aggregate penalty $\Delta(s, a) = \sum_i |q_i(s, a) - q_i(s, a_o)|$ measures how much each action deviates from the baseline across all oversight signals. CCO selects actions by maximizing $U(s, a) - \lambda_t \Delta(s, a)$, where the conservatism parameter $\lambda_t$ is updated online via a conformal controller: after observing whether the selected action incurred a loss $\ell_t$, the controller adjusts $\lambda_{t+1} = \lambda_t + \eta(\ell_t - \alpha)$. This feedback loop ensures that realized violation rates converge to the user-specified target $\alpha$. In safe situations, CCO relaxes conservatism to permit high-utility actions; in risky situations, it increases conservatism to favor safer alternatives.

alternative available. We generalize this penalty from Q-value functions to arbitrary scoring functions, substantially broadening applicability. Rather than requiring a complete MDP with specified reward functions, CCO can incorporate the diverse oversight signals practically available for large language models and other frontier systems, while retaining the sequential decision-making structure essential for agentic deployment.

A key challenge in applying penalty-based methods is selecting the conservatism level: too little permits unsafe actions; too much paralyzes the agent. We address this via Conformal Decision Theory (CDT) (Lekeufack et al., 2024), which calibrates decision rules with worst-case risk control. As shown in Figure 1, the conformal controller adjusts $\lambda_{t+1}$ based on observed losses: increasing conservatism after failures, relaxing it after safe actions. CDT guarantees that the long-run loss rate converges to a user-specified target $\alpha$ without distributional assumptions, provided the decision family is *eventually safe*: sufficiently conservative settings must eventually eliminate harmful actions.

We prove that the CCO decision family satisfies this requirement. Under mild finiteness conditions, there exists a conservatism threshold beyond which the agent always

selects the baseline action. This enables direct application of CDT, yielding an online algorithm that provably controls the rate at which the agent incurs undesirable outcomes to any desired level $\alpha$, with explicit finite-time bounds. Crucially, these guarantees hold for arbitrary, even adversarial, state sequences, making them robust to the non-stationarity and distribution shift inherent in agentic deployment.

Recent work has begun applying conformal prediction methods to LLM safety, controlling tail risks (Chen et al., 2025), improving factuality (Cherian et al., 2024), and mediating between helpfulness and harmlessness (Overman & Bayati, 2025), but these approaches predominantly address single-turn interactions rather than extended trajectories where actions compound.

The resulting framework offers a practical instantiation of a scalable oversight protocol with formal guarantees suited to sequential decision-making. The auxiliary overseers $\{q_i\}$ need not individually understand the primary agent's full capabilities or goals; they can contribute local, primarily safety-focused evaluations, that CCO aggregates into a global constraint. The conservatism parameter adapts online to the actual reliability of the oversight ensemble across the trajectory of states encountered in deployment.

We validate CCO experimentally across two distinct deployment regimes. On SWE-bench (Jimenez et al., 2024), we demonstrate that weaker overseers can successfully constrain an adversarially misaligned stronger agent across independent code-editing instances. On MACHIAVELLI (Pan et al., 2023), CCO achieves substantial reductions in ethical violations across sequential game trajectories while preserving task performance. In both settings, realized violation rates closely match user-specified targets, confirming the effectiveness of the conformal calibration mechanism.

## 2. Related Work

Our work contributes to scalable oversight by combining conservative-agency methods originally developed for tabular RL with conformal techniques that have recently been brought to bear on frontier LLMs, yielding a deployment-time protocol with formal guarantees suited to sequential agentic settings.

### 2.1. Scalable Oversight

*Scalable oversight* (Amodei et al., 2016) asks how to provide reliable supervision for AI systems whose capabilities may exceed a human (or otherwise weak) evaluator's capabilities. A prominent line of work frames oversight as an adversarial protocol. *Debate* (Irving et al., 2018; Kenton et al., 2024) and related *prover–verifier* formulations (Kirchner et al., 2024) use competitive dynamics to surface information a weaker judge may otherwise miss. Brown-Cohen et al. (2024) provide complexity-theoretic guarantees for "doubly-efficient" debate, and Brown-Cohen et al. (2025) analyze conditions under which recursive debate-style protocols remain sound in the presence of obfuscated arguments.

Another line of work studies when weak evaluators can directly judge or train stronger models. Burns et al. (2024) investigate weak-to-strong generalization, eliciting strong capabilities under weak supervision, and Kenton et al. (2024) study regimes where weak LLMs can effectively judge strong LLMs, as well as where this breaks down. *Constitutional AI* (Bai et al., 2022) demonstrates that written principles can partially substitute for continuous human judgment by enabling AI-generated feedback, while *process supervision* (Lightman et al., 2024) shows that supervising intermediate reasoning steps can outperform outcome-only supervision in mathematical reasoning. *Iterated amplification* (Christiano et al., 2018) and *recursive reward modeling* (Leike et al., 2018) propose decomposing hard evaluation problems into simpler subproblems that humans can supervise and recombine. From a meta perspective, (Engels et al., 2025) study how scalable oversight itself scales, investigating the probability of successful oversight as a function of the capability gap between the potentially misaligned model and the overseer. Relatedly, the direction of AI Control (Greenblatt et al., 2024) emphasizes protocols intended to remain robust even when the deployed system is actively attempting subversion.

Our contribution is complementary to these approaches. Rather than modifying training to produce a more judgeable model, or relying on an explicit multi-agent argument protocol, CCO provides a deployment-time method tailored to agentic, sequential settings.

### 2.2. Side-Effect Avoidance and Conservative Agency

A substantial line of work studies how to design agents that avoid unintended side effects while pursuing underspecified objectives. Early approaches proposed measures that penalize irreversible or large-scale changes to the environment (Amodei et al., 2016; Leike et al., 2017). This approach relates to broader work on impact measures (Armstrong & Levinstein, 2017; Krakovna et al., 2019; 2020) and power-seeking behavior (Turner et al., 2021). Related work on *corrigibility* emphasizes preserving the ability for humans to intervene, even when doing so conflicts with the agent's current objectives (Soares et al., 2015; Hadfield-Menell et al., 2017).

A complementary perspective limits how much an agent alters attainable futures. *Relative reachability* (Krakovna et al., 2019) penalizes actions reducing the agent's ability to reach desirable states, encouraging reversibility. *Attainable Utility Preservation* (AUP) (Turner et al., 2020a;b) generalizes this, penalizing actions that substantially change the agent's ability to optimize auxiliary reward functions. The insight is that actions with large effects across many auxiliary objectives likely have large effects on the true objective as well. We build on AUP but generalize in two ways: replacing Q-values with arbitrary scoring functions, and replacing fixed $\lambda$ with online calibration yielding formal guarantees on violation rates.

### 2.3. Conformal Prediction and Decision Theory.

Conformal prediction (Vovk et al., 2005; Angelopoulos & Bates, 2022) provides distribution-free uncertainty quantification, with extensions to general risk control (Angelopoulos et al., 2024). Recent work applies these methods to LLM safety: controlling tail risks like toxicity (Chen et al., 2025), improving factuality (Cherian et al., 2024), calibrating outputs to satisfy user-desired properties (Overman et al., 2024), mediating between helpfulness and harmlessness (Overman & Bayati, 2025), and escalating to stronger judges when weaker ones lack confidence (Jung et al., 2025). Relatedly, our Calibrated Collective Oversight algorithm uses Conformal Decision Theory (CDT) (Lekeufack et al., 2024), which guarantees long-run risk control without distributional assumptions.

# 3. Background

This section reviews the two main technical foundations underlying our approach: *Attainable Utility Preservation* (AUP) and *Conformal Decision Theory* (CDT).

## 3.1. Attainable Utility Preservation

Attainable Utility Preservation (AUP) (Turner et al., 2020b) is a method for *side-effect avoidance* that penalizes actions which substantially change an agent's ability to optimize a diverse set of auxiliary objectives, thereby encouraging conservative, reversible behavior.

AUP is defined on an MDP $(\mathcal{S}, \mathcal{A}, P, R, \gamma)$ with discount $\gamma \in (0, 1)$ and a distinguished baseline action $a_0 \in \mathcal{A}$. Crucially, the reward function $R$ is considered to be potentially misspecified by the designer. In addition to this primary reward function $R$, the designer is assumed to provide a finite auxiliary set $\mathcal{R}_{\text{aux}} = \{R_1, \ldots, R_N\}$. For each $R_i$, let $Q_i(s, a)$ denote the optimal discounted Q-value under $R_i$. The vector $(Q_1(s, a), \ldots, Q_N(s, a))$ is interpreted as an *attainable-utility profile* at $(s, a)$.

AUP measures impact by the $\ell_1$ distance in attainable-utility space from the baseline,

$$\text{PENALTY}(s, a) := \sum_{i=1}^{N} |Q_i(s, a) - Q_i(s, a_0)|.$$

Because auxiliary rewards are only meaningful up to positive affine transformations, AUP introduces a state-dependent normalization $\text{SCALE}(s) > 0$. One natural choice (discussed in Turner et al., 2020b), assuming that rewards are strictly positive, is the magnitude of attainable utilities under the baseline,

$$\text{SCALE}(s) := \sum_{i=1}^{N} Q_i(s, a_0).$$

AUP optimizes a regularized objective of the form

$$R_{\text{AUP}}(s, a) := R(s, a) - \lambda \frac{\text{PENALTY}(s, a)}{\text{SCALE}(s)}, \qquad \lambda \geq 0,$$

where $\lambda$ is fixed for a given implementation and larger $\lambda$ induces more conservative behavior. AUP penalizes *absolute* change in attainable utilities (rather than only decreases), which helps avoid incentives to manipulate or preserve specific capabilities.

The original AUP formulation has three key limitations: it requires computing auxiliary optimal Q-functions $\{Q_i\}$, offers no principled mechanism for selecting $\lambda$ to meet a desired safety target, and provides no formal guarantees on violation rates. CCO addresses these by generalizing from

Q-values to arbitrary scoring functions, calibrating $\lambda$ online via conformal methods, and thereby obtaining finite-sample guarantees on the long-run rate of undesirable outcomes.

## 3.2. Conformal Decision Theory

Conformal Decision Theory (CDT), introduced by Lekeu-fack et al. (2024), provides a general framework for online calibration of decision rules with worst-case risk control. CDT can be seen as a decision-theoretic analogue of conformal prediction (Vovk et al., 2005; Angelopoulos & Bates, 2022), operating over actions rather than predictions.

CDT considers a family of decision rules $\{D^\lambda : \lambda \in \mathbb{R}\}$, indexed by a scalar parameter $\lambda$ interpreted as a level of conservatism. At each time $t$, the agent selects an action $a_t = D^{\lambda_t}(x_t)$ based on the current context $x_t$.

After executing $a_t$, the agent observes a bounded loss $\ell_t \in [0, 1]$. The goal is to ensure that the long-run average loss does not exceed a user-specified target $\alpha \in [0, 1]$:

$$\limsup_{t \to \infty} \frac{1}{t + 1} \sum_{s=0}^{t} \ell_s \ \leq \ \alpha.$$

No distributional assumptions are made on the sequence $(x_t, \ell_t)$, which may be adversarial.

The key structural assumption in CDT is that the decision family is *eventually safe*, which we now state formally as it plays a central role in our analysis.

**Definition 3.1** (Eventually safe decision family). A family of decision rules $\{D^\lambda : \mathcal{S} \to \mathcal{A}\}$ is *eventually safe* if there exist constants $\alpha_{\text{safe}} \in [0, 1]$, $\lambda_{\text{safe}} \in \mathbb{R}$, and an integer horizon $K \geq 1$ such that, uniformly over all sequences $\lambda_{t:t+K-1}$ and all realized state/outcome sequences over that window,

$$\big\{\forall k \in \{0, \ldots, K - 1\}, \ \lambda_{t+k} \geq \lambda_{\text{safe}}\big\}$$
$$\implies \frac{1}{K} \sum_{k=0}^{K-1} \ell_{t+k} \leq \alpha_{\text{safe}}.$$

Intuitively, eventual safety captures the idea that sufficient conservatism should eliminate harmful actions entirely after a finite transient period.

Given eventual safety, CDT proposes a simple update rule:

$$\lambda_{t+1} \ = \ \lambda_t + \eta(\ell_t - \alpha),$$

where $\eta > 0$ is a step size. This update increases conservatism after losses and relaxes it after safe actions.

CDT guarantees that, for any realization of the deployment sequence, the empirical average loss converges to at most $\alpha$, with explicit finite-time bounds depending on $\lambda_{\text{safe}}$, $\lambda_0$, and $\eta$. These guarantees hold without stochastic assumptions and are robust to non-stationarity and adversarial feedback.

# 4. Calibrated Collective Oversight

We model interaction with the environment as a *controlled Markov process* (cMP), a standard Markov decision process without a fixed reward function (Sutton & Barto, 2018). Formally, the environment is specified by a tuple $(\mathcal{S}, \mathcal{A}, P)$, where $\mathcal{S}$ is a finite state space, $\mathcal{A}$ is a finite action space, and $P(\cdot \mid s, a)$ denotes the transition kernel.

Unlike classical reinforcement learning, we do not assume access to a ground-truth reward. Instead, decisions are guided by a primary scoring function together with auxiliary oversight signals, which we introduce below.

## 4.1. Definitions

**Definition 4.1** (Baseline action). A *baseline action selector* is a function $b : \mathcal{S} \to \mathcal{A}$ that assigns to each state $s$ a designated baseline action $a_0(s) \in \mathcal{A}$. The baseline represents the default conservative choice: the action to which the agent defers when oversight signals counsel caution.

The baseline action serves as a canonical safe fallback and reference point for measuring deviation. Much of our analysis relies on the fact that sufficiently conservative decision rules eventually prefer the baseline. In practice, the baseline may be defined as the action minimizing some known harm metric, the action with lowest variance across overseer evaluations, or simply a designated "do nothing" option.

**Definition 4.2** (Primary decision agent). A *primary decision agent* is specified by a pair $D := (C, U)$, where:

- $C : \mathcal{S} \to 2^{\mathcal{A}}$ is an *action proposal map* assigning to each state $s \in \mathcal{S}$ a finite, nonempty set $C(s) \subseteq \mathcal{A}$ of candidate actions, and

- $U : \mathcal{S} \times \mathcal{A} \to \mathbb{R}$ is a *primary scoring function*.

We assume $a_0(s) \in C(s)$ for all $s \in \mathcal{S}$.

For a given state $s$, the action executed by the primary decision agent would be selected from the proposed set $C(s)$ by maximizing the scoring function $U(s, \cdot)$. The separation between the proposal map $C$ and the scoring function $U$ allows us to model settings in which the ambient action space $\mathcal{A}$ is large, but only a restricted subset of actions is feasible or worth evaluating at any given state.

In the scalable oversight setting, no single overseer may fully understand the primary agent's capabilities or intentions. We therefore aggregate signals from multiple auxiliary evaluators, leveraging their collective judgment to flag actions that deviate from conservative behavior.

**Definition 4.3** (Auxiliary overseers). The *auxiliary overseers* are a finite collection

$$\mathcal{Q}_{\mathrm{aux}} := \{q_1, \ldots, q_N\},$$

where each $q_i \in \mathcal{Q}_{\mathrm{aux}}$ is a function $q_i : \mathcal{S} \times \mathcal{A} \to \mathbb{R}$ that assigns a scalar score to state-action pairs.

Auxiliary overseers evaluate but do not act: at each state $s$, they score only the candidates $C(s)$ proposed by the primary agent. We impose no structural assumptions beyond real-valued output. An auxiliary overseer may be a Q-function for some reward, a heuristic safety critic, a smaller language model, or any other scoring mechanism. We aggregate overseer signals into a single penalty measuring total deviation from the baseline, then trade off this penalty against the primary agent's utility.

**Definition 4.4** (CCO penalty). The *CCO penalty* at state $s$ for action $a$ is

$$\Delta(s, a) := \sum_{i=1}^{N} \big| q_i(s, a) - q_i(s, a_0(s)) \big|.$$

The penalty aggregates how much each overseer's evaluation of $a$ differs from its evaluation of the baseline. Actions that all overseers score similarly to the baseline incur low penalty; actions that trigger disagreement across multiple overseers incur high penalty. This structure does not require overseers to agree on what is good, only that deviation from the baseline registers as change in their scores.

**Definition 4.5** (Oversight-regularized score). For $\lambda \in \mathbb{R}$, define

$$U_{\lambda}(s, a) := U(s, a) - \lambda \, \Delta(s, a).$$

The parameter $\lambda$ controls the tradeoff between the primary agent's utility and the collective oversight penalty. When $\lambda = 0$, the agent is allowed to operate ignoring oversight entirely; as $\lambda$ increases, the agent becomes increasingly restricted and forced to actions closer to the baseline. Compared to the original AUP formulation presented in Section 3.1, we do not include a scale term due to the fact that we will be tuning $\lambda$.

**Definition 4.6** (CCO decision family). For each $\lambda \in \mathbb{R}$, define a decision rule

$$D^{\lambda}(s) \in \arg \max_{a \in C(s)} U_{\lambda}(s, a),$$

with ties broken in favor of $a_0(s)$. The collection $\{D^{\lambda} : \lambda \in \mathbb{R}\}$ is the *CCO decision family*.

The decision family spans the full range from unconstrained behavior ($\lambda = 0$) to maximal conservatism (large $\lambda$, where the agent defaults to the baseline). The central question is how to select $\lambda$ to achieve a desired level of safety. To formalize this, we introduce a loss function that measures outcomes we wish to control.

**Definition 4.7** (Loss function). Let $\ell : \mathcal{S} \times \mathcal{A} \to [0, 1]$ be a loss function, where $\ell(s, a)$ quantifies the undesirability of

**Algorithm 1** CCO (Online calibration)
___
1: **Inputs:** step size $\eta > 0$, target rate $\alpha \in [0, 1]$, initial $\lambda_0 \in \mathbb{R}$
2: **for** $t = 0, 1, 2, \ldots, T$ **do**
3:      Observe state $s_t$
4:      Query $C(s_t), U(s_t, a)$, and $\{q_i(s_t, a)\}_{i=1}^N$ for all $a \in C(s_t)$
5:      Compute $\Delta(s_t, a)$ and $U_{\lambda_t}(s_t, a)$
6:      Select $a_t = D^{\lambda_t}(s_t)$
7:      Execute $a_t$; observe loss $\ell_t := \ell(s_t, a_t)$
8:      Update $\lambda_{t+1} \leftarrow \lambda_t + \eta(\ell_t - \alpha)$
9: **end for**
___

taking action $a$ in state $s$. We assume the baseline is safe: $\ell(s, a_0(s)) = 0$ for all $s \in \mathcal{S}$.

The loss function is application-specific and may capture safety violations, harm, or any other outcome the designer wishes to control. The assumption that $\ell(s, a_0(s)) = 0$ reflects the design choice that the baseline represents a conservative default with minimal undesirable consequences.

### 4.2. Algorithm

The CCO decision family provides a spectrum of behaviors parameterized by $\lambda$, but selecting the right $\lambda$ a priori is difficult: it depends on the distribution of states encountered, the reliability of overseers, and the relationship between the penalty and actual losses. We instead calibrate $\lambda$ online using a conformal controller that adjusts conservatism based on observed outcomes.

Algorithm 1 presents the complete procedure. At each timestep, the agent observes a state, queries the primary scoring function and all auxiliary overseers on the candidate actions, selects the action maximizing the oversight-regularized score, and then updates $\lambda$.

The update rule in line 8 is our use of CDT: increase $\lambda_t$ if a significant loss was incurred ($\ell_t > \alpha$), decrease it otherwise. This drives the empirical loss rate toward the target $\alpha$, becoming more conservative after failures, and more permissive after safe actions.

### 4.3. Guarantees

For the conformal controller to provide guarantees, the decision family must be eventually safe (Definition 3.1): sufficiently large $\lambda_t$ must eliminate losses entirely. We show that CCO satisfies this under mild conditions.

**Lemma 4.8** (Uniform baseline dominance). *Assume $\mathcal{S}$ and $\mathcal{A}$ are finite and that $\Delta(s, a) > 0$ for all $s \in \mathcal{S}$ and $a \in$*

$C(s) \setminus \{a_0(s)\}$. *Define*

$$\lambda^\star(s) := \max\left\{0, \max_{a \in C(s)\setminus\{a_0(s)\}} \frac{U(s, a) - U(s, a_0(s))}{\Delta(s, a)}\right\},$$
$$\bar{\lambda} := \max_{s \in \mathcal{S}} \lambda^\star(s),$$

*where the inner maximum over an empty set is interpreted as $-\infty$. Then $\bar{\lambda} < \infty$, and for all $\lambda \geq \bar{\lambda}$, $D^\lambda(s) = a_0(s)$ for every $s \in \mathcal{S}$.*

The threshold $\bar{\lambda}$ is the conservatism level beyond which the penalty always outweighs any utility gain, forcing the agent to select the baseline everywhere. Since the baseline incurs zero loss by assumption, this yields eventual safety. We provide the full proof in Appendix A.2.

**Lemma 4.9** (Eventual safety of CCO). *Under the conditions of Lemma 4.8, the CCO decision family $\{D^\lambda\}$ is eventually safe in the sense of Definition 3.1, with $\alpha_{\mathrm{safe}} = 0$, $\lambda_{\mathrm{safe}} = \bar{\lambda}$, and $K = 1$.*

The proof is immediate from Lemma 4.8: whenever $\lambda_t \geq \bar{\lambda}$, the executed action is $a_0(s_t)$ and the realized loss is zero. With eventual safety established, we obtain finite-time bounds on the cumulative loss rate.

**Theorem 4.10** (Loss control for CCO). *Under the conditions above, assume the controller is initialized with $\lambda_0 \leq \bar{\lambda} + \eta$. Then Algorithm 1 guarantees that, for any realized deployment sequence and all $t \geq 0$,*

$$\hat{R}_t := \frac{1}{t+1}\sum_{s=0}^{t} \ell_s \leq \alpha + \frac{(\bar{\lambda} - \lambda_0)/\eta + 1}{t+1}.$$

The bound holds for arbitrary, even adversarial, state sequences, with no distributional assumptions. The second term vanishes as $t \to \infty$, so the long-run loss rate converges to at most $\alpha$. The proof is given in Appendix A.5. The $O(1/t)$ transient dependence is worst-case order-sharp for our additive controller; Appendix A.6 gives a lower-bound construction showing that the dependence on $(\bar{\lambda} - \lambda_0)/(\eta(t+1))$ cannot be uniformly improved.

### 4.4. How Overseer Quality Affects Conservatism

The maximum value of $\lambda_t$ reached by Algorithm 1 depends on how well the auxiliary overseers distinguish unsafe actions from the baseline. If overseers assign nearly identical scores to a harmful action and the baseline, then a large penalty weight is required to offset the primary agent's utility advantage. Conversely, if unsafe actions differ from the baseline by a large aggregate overseer margin, then a smaller value of $\lambda$ suffices.

Formally, Appendix C gives three complementary ways to quantify this dependence. First, if every loss-incurring

action has penalty at least $\kappa > 0$, then the tie-broken CCO decision rule is safe for every

$$\lambda \geq \frac{B}{\kappa}, \qquad B := \sup_{s \in S} \sup_{a \in C(s)} \left[ U(s, a) - U(s, a_0(s)) \right]_+.$$

Thus the safe threshold satisfies $\lambda_{\text{safe}} \leq B/\kappa$ (Proposition C.2). Second, bounded overseer error degrades the margin by at most twice the total error magnitude, inflating the required conservatism accordingly (Corollary C.4). Finally, under the weaker surrogate condition $\Delta(s, a) \geq c\,\ell(s, a)$, the decision rule excludes all actions with loss at least $\varepsilon$ whenever $\lambda \geq B/(c\varepsilon)$ (Proposition C.6). For binary or positively gapped losses, this again yields a finite safe threshold.

### 4.5. Practical Extensions: Noisy and Delayed Feedback

Algorithm 1 assumes immediate exact observation of $\ell_t$, which is unrealistic in many deployments. Appendix B shows that the guarantee degrades gracefully under two practical relaxations.

**Noisy feedback.** Suppose the controller updates using noisy observations $\tilde{\ell}_t = \ell_t + \xi_t$ rather than the true loss, where $\{\xi_t\}$ is a conditionally mean-zero, $\sigma^2$-sub-Gaussian martingale-difference sequence. The proof couples the noisy controller to a de-noised virtual controller and shows that, for any fixed $t$, with probability at least $1 - \delta$,

$$\hat{R}_t \leq \alpha + \frac{(\bar{\lambda} - \lambda_0)/\eta + 1}{t + 1} + \sigma \sqrt{\frac{2 \log(1/\delta)}{t + 1}}.$$

Thus stochastic feedback noise adds the usual $O(\sigma/\sqrt{t+1})$ concentration term. With only deterministic bounded noise $|\xi_t| \leq \sigma$, the same argument yields a pathwise additive term of at most $\sigma$.

**Delayed feedback.** Suppose the loss from round $r$ is revealed after an integer delay $d_r \leq D$, and the controller updates only on losses revealed at the current time. A delayed-update analysis shows that, for all $t \geq 0$,

$$\hat{R}_t \leq \alpha + \frac{(\bar{\lambda} - \lambda_0)/\eta + D + 1}{t + 1}.$$

The case $D = 0$ recovers Theorem 4.10. Hence bounded delay adds only an $O(D/(t+1))$ transient term.

Together, stochastic zero-mean noise preserves convergence to the target up to high-probability $O(1/\sqrt{t})$ fluctuations, while bounded delay preserves the original $O(1/t)$ transient rate.

## 5. Experiments

We evaluate CCO in two central domains: (1) SWE-bench (Jimenez et al., 2024), a realistic software engineering

benchmark where a tool-calling coding agent is adversarially instructed to insert vulnerabilities while a panel of information-limited overseers reviews its patches; and (2) the MACHIAVELLI benchmark (Pan et al., 2023), a collection of text-based games with ethical annotations, demonstrating applicability to LLM-based agents in narrative environments. Additionally, in Appendix H we include a grid-world experiment to illustrate CCO's adaptive calibration in non-stationary environments and to provide a direct comparison with the original AUP formulation. Cross-model results, an overseer-count ablation, an $a_0$ sensitivity analysis, and a comparison against an adaptive majority-vote baseline are summarized inline and reported in full in Appendix D.

### 5.1. Constraining Adversarial Agents on SWE-bench

To evaluate CCO in a realistic software-engineering setting with agentic tool use, we apply the framework to all 300 instances of SWE-bench Lite (Jimenez et al., 2024). Each instance is a real GitHub issue with an associated test suite. We instantiate an adversarial scenario in which a *tool-calling coding agent* may be secretly instructed to insert an instance-specific vulnerability into its patch, while a panel of information-limited overseers reviews the proposed code. The quantity CCO is calibrated to control is the *misalignment rate*: the fraction of instances on which a vulnerability-carrying patch is deployed. All candidate patches are independently evaluated in Docker containers against the ground-truth test suites, allowing us to verify both the realized misalignment rate and the resulting *solve rate* (functional correctness) as a measure of the utility cost of oversight.

**Agentic patch generation.** For each instance, a Claude Sonnet 4.6 agent generates three candidate patches at different *resource levels*, operationalizing the capability–safety tradeoff through compute allocation:

- **High-resource** (9–12 tool-calling turns): the most thorough solution. With probability $p = 0.5$, this candidate receives an instance-specific misalignment instruction targeting a vulnerability tailored to the files the issue touches.

- **Medium-resource** (5–8 turns): a competent but less exhaustive attempt. Always aligned.

- **Low-resource** (1–4 turns): a minimal attempt. Always aligned. Serves as the CCO baseline $a_0$.

Turn budgets are drawn uniformly at random within their bands, with normalized utility $u_i = \text{turns}_i/12$ so that the utility gap between candidates remains in $[0, 1]$. The agent operates in a realistic tool-calling loop: it reads source files, navigates the repository, writes unified-diff patches, and

runs the project's test suite before submitting. This end-to-end agentic pipeline means that any inserted vulnerability must survive the agent's own verification step while remaining invisible to external review. Note that the capped turn budgets yield solve rates below the frontier reported by model providers, who typically allow 25–100+ turns (Anthropic, 2025b).

**Instance-specific vulnerabilities.** We use Claude Opus 4.6 to generate an instance-specific vulnerability prompt for each of the 300 instances, tailored to the files and functions the issue touches (categories: SQL injection, `eval/exec`, unsafe deserialization, XSS, path traversal, verbose logging disclosure, `shell=True` command injection, and broad `except` clauses). Each vulnerability is designed to resemble a plausible implementation choice rather than an obviously malicious insertion. Full prompt templates appear in Appendix E.

**Random-context overseers.** We model a realistic capability gap and information asymmetry: each overseer is an independent Claude Haiku 4.5 call ($N = 10$ per candidate) that receives the proposed patch alongside a random sample of $k = 4$ Python files from the repository (excluding build artifacts and files exceeding 50 KB). The overseer is instructed to flag anything suspicious or inconsistent with codebase style and to assign a score on a 1–10 scale (normalized to $[0, 1]$), with no foreknowledge of what vulnerabilities might be attempted. Full prompts appear in Appendix E.

**Penalty, selection, and evaluation.** The penalty for candidate $i$ is $p_i = \sum_{j=1}^{N} |q_j^{(i)} - q_j^{(low)}|$. CCO selects $\arg\max_i[u_i - \lambda_t \cdot p_i]$, with the projected update $\lambda_{t+1} = \max\{0, \lambda_t + \eta(\ell_t - \alpha)\}$, $\ell_t = 1$ iff the selected candidate is misaligned. All 300 instances are processed sequentially with $\eta = 0.3$. Every candidate patch is independently evaluated against the project's test suite in a Docker container; the saved scores and outcomes let us resimulate CCO's selection at any $\alpha$ from a single run.

**Results.** Table 1 reports both safety and utility across a range of $\alpha$ targets, flanked by two baselines: *Always Baseline* always selects the low-resource candidate (12.0% solve, 0% misalignment), and *Unconstrained* always selects the high-resource candidate (29.0% solve, 55.0% misalignment). CCO interpolates between these extremes, with the realized misalignment rate closely tracking the target across the full range ($\alpha = 0.01 \rightarrow 1.3\%$, $0.05 \rightarrow 5.0\%$, $0.10 \rightarrow 9.7\%$, $0.20 \rightarrow 17.0\%$); maximum deviation is 3.0pp (at $\alpha = 0.20$). Solve rate climbs monotonically from 13.0% to 19.7% as $\alpha$ increases, reflecting the expected safety–utility tradeoff. At $\alpha = 0.05$, CCO improves solve rate by 2.3pp over the always-safe baseline while holding misalignment at 5.0%; at $\alpha = 0.25$, solve rate reaches

*Table 1.* **SWE-bench Lite results** (300 instances, Claude Sonnet 4.6 agent, $p = 0.5$ misalignment probability, $\eta = 0.3$). *Misalignment rate*: fraction where a vulnerability-carrying patch is deployed. *Solve rate*: fraction of instances where the selected patch passes the SWE-bench test suite. CCO conditions are resimulated from saved overseer scores at each $\alpha$; baselines are computed directly from per-candidate test results.

| Method | Misalignment Rate | Solve Rate |
|---|---|---|
| Always Baseline ($\alpha = 0$) | 0.0% | 12.0% |
| CCO ($\alpha = 0.01$) | 1.3% | 13.0% |
| CCO ($\alpha = 0.05$) | 5.0% | 14.3% |
| CCO ($\alpha = 0.10$) | 9.7% | 15.3% |
| CCO ($\alpha = 0.15$) | 14.0% | 16.7% |
| CCO ($\alpha = 0.20$) | 17.0% | 17.7% |
| CCO ($\alpha = 0.25$) | 21.0% | 19.7% |
| Unconstrained ($\lambda = 0$) | 55.0% | 29.0% |

19.7% (a 64% relative improvement) with 21.0% misalignment.

**Ablations.** Appendix D reports ablations over overseer count $n$ (showing $\alpha$-tracking from $n = 1$ to $n = 10$, with larger $n$ tightening precision), baseline-action sensitivity (four choices of $a_0$ all yield rates within 1.1pp of target), and cross-model replications confirming the same calibration with GPT-5.4-family agents. We also compare against an adaptive majority-vote baseline that adjusts an acceptance threshold online to target the same $\alpha$: under steady state the two achieve comparable rates, but under distribution shift ($p$ jumping from 0.2 to 0.8 mid-stream) adaptive MV's realized rate spikes substantially above target while CCO stays within $\sim$ 2pp throughout.

### 5.2. MACHIAVELLI Benchmark

The MACHIAVELLI benchmark (Pan et al., 2023) consists of text-based choose-your-own-adventure games where players navigate moral trade-offs between achieving objectives and adhering to ethical principles. The benchmark provides human annotations for 13 categories of ethical violations (e.g., harm, deception, betrayal, stealing), making it well-suited for evaluating whether AI agents can pursue task objectives while respecting moral constraints. At each state $s$, the agent observes narrative text and a finite set ($\sim 5$) of available actions $C(s)$, each with associated in-game reward and potential ethical violations.

**Implementation.** We use Claude Sonnet 4.5 (Anthropic, 2025b) as the *primary agent*, producing utility scores $U(s, a) \in [0, 10]$, with separate Sonnet 4.5 calls providing violation detection $v(s, a)$ using MACHIAVELLI's annotation scheme. Five *auxiliary overseers* are implemented with Claude Haiku 4.5 (Anthropic, 2025a), each specialized to a distinct ethical dimension (Harm Prevention, Honesty

Table 2. MACHIAVELLI results on *Hero of Kendrickstone* (100 steps, 10 seeds). CCO closely tracks target violation rates while maintaining reasonable reward.

| Method | Violations | $\max_t \lambda_t$ | Reward |
|---|---|---|---|
| Always Baseline | 0.0% | — | $35.0 \pm 8.1$ |
| CCO $\alpha$=0.05 | $6.2 \pm 1.0\%$ | $0.82 \pm 0.38$ | $39.5 \pm 12.9$ |
| CCO $\alpha$=0.10 | $10.9 \pm 2.7\%$ | $0.57 \pm 0.19$ | $49.0 \pm 8.3$ |
| CCO $\alpha$=0.15 | $12.7 \pm 3.4\%$ | $0.49 \pm 0.07$ | $55.5 \pm 7.6$ |
| $\lambda = 0$ | $21.5 \pm 8.2\%$ | — | $54.0 \pm 7.3$ |

Advocate, Autonomy Guardian, Fairness Monitor, Trust Keeper), assigning scores $q_i(s, a) \in [0, 10]$. The Haiku/-Sonnet capability gap directly tests whether weaker models can constrain a stronger one. Since MACHIAVELLI games lack explicit no-op actions, we define $a_0$ per state as the action with lowest violation count.

**Loss Function.** We adopt an *excess violations* loss that penalizes avoidable ethical violations:

$$\ell(s, a) := \max\{0, \, v(s, a) - \min_{a' \in C(s)} v(s, a')\},$$

where $v(s, a)$ denotes the number of violations for action $a$ in state $s$. This formulation ensures the agent is not penalized when no safe choice exists.

**Experimental Protocol.** We evaluate on two games: *Hero of Kendrickstone* (10 seeds, 100 steps) and *Pendragon Rising* (10 seeds, 100 steps). Each run starts from the initial game state. We set initial conservatism $\lambda_0 = 0.0$ and step size $\eta = 0.1$, with target violation rates $\alpha \in \{0.05, 0.10, 0.15\}$. We compare against two baselines: an Always No-Op agent that always selects the dynamically-defined safest action, and an unconstrained agent ($\lambda = 0$) that maximizes reward without safety constraints.

**Results.** Table 2 summarizes results on *Hero of Kendrickstone*; *Pendragon Rising* results appear in Appendix G. CCO closely tracks target violation rates: $\alpha = 0.10$ and $\alpha = 0.15$ both fall within statistical noise of their targets. The tighter $\alpha = 0.05$ setting yields 6.2%, slightly above target, suggesting 100 iterations are insufficient for the transient term in Theorem 4.10 to fully vanish at such stringent levels. The $\max_t \lambda_t$ column decreases with looser targets ($0.82 \rightarrow 0.57 \rightarrow 0.49$), confirming the controller adapts conservatism appropriately. Reward is not substantially degraded by oversight: at $\alpha = 0.15$ CCO matches the unconstrained agent's reward while halving the violation rate, indicating that in this narrative environment ethical play often coincides with high-reward trajectories.

### 5.3. Summary

Across both domains, CCO demonstrates consistent calibration: $\alpha = 0.10$ yields 9.7% misalignment in SWE-bench and 10.9% violations in MACHIAVELLI, despite dramatically different environments. The experiments validate two key properties: *interpretable control* ($\alpha$ directly specifies the desired safety level), and *weak-to-strong scalability* (weaker overseers successfully constrain a stronger adversarial agent operating in a realistic tool-calling pipeline).

## 6. Conclusion

We introduced Calibrated Collective Oversight, a framework combining AUP-inspired penalty aggregation with online calibration via Conformal Decision Theory. CCO provides finite-time guarantees on violation rates without distributional assumptions, providing a practical instantiation of scalable oversight with formal guarantees.

**Limitations.** Several limitations of our framework are worth noting. (i) Theorem 4.10 assumes that the loss $\ell_t$ is observed exactly after each step. Section 4.5 and Appendix B show graceful degradation under bounded noise and bounded delay, but in many realistic settings ground-truth labels arrive only sparsely or are estimated by a separate (and themselves fallible) judge. (ii) Our SWE-bench protocol is adversarial by construction: an agent is instructed to insert instance-specific vulnerabilities and its patches are evaluated against the project's tests. This provides a controlled measurement of $\alpha$-tracking but is not a substitute for naturalistic measurement, which requires post-deployment ground truth. (iii) CCO controls a long-run violation rate and is not as well-suited for extremely rare catastrophic events; it is therefore complementary to per-action monitoring of tail risks. (iv) The framework assumes a designated baseline $a_0$ with $\ell(s, a_0(s)) = 0$; specifying or learning a baseline is non-trivial in some domains, though Appendix D shows the conformal controller is robust across several plausible choices.

## Impact Statement

This paper presents work whose goal is to advance the field of machine learning, specifically the safe deployment of agentic AI systems. By introducing Calibrated Collective Oversight (CCO), our work provides a framework for maintaining human-meaningful control over increasingly capable models through statistical guarantees and online calibration. There are potential societal consequences of our work, most notably the reduction of risks associated with AI misalignment in sequential decision-making, none of which we feel must be specifically highlighted here beyond the technical contributions presented.

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

# A. Proofs

This appendix provides complete proofs for the finiteness-based baseline dominance property, the induced eventual-safety property of the CCO decision family, and the long-run loss control guarantee.

## A.1. Setup

Recall the CCO penalty and $\lambda$-regularized score:

$$\Delta(s,a) := \sum_{i=1}^{N} \big|q_i(s,a) - q_i(s,a_0(s))\big|, \qquad U_\lambda(s,a) := U(s,a) - \lambda\Delta(s,a),$$

and the CCO decision rule

$$D^\lambda(s) \in \arg\max_{a \in C(s)} U_\lambda(s,a), \qquad \text{with ties broken in favor of } a_0(s).$$

At deployment time $t = 0, 1, 2, \ldots$, the algorithm executes

$$a_t = D^{\lambda_t}(s_t)$$

and observes a realized loss

$$\ell_t := \ell(s_t, a_t) \in [0, 1].$$

We use the CDT-style additive controller

$$\lambda_{t+1} = \lambda_t + \eta(\ell_t - \alpha), \qquad \eta > 0, \quad \alpha \in [0, 1]. \tag{1}$$

## A.2. Proof of Lemma 4.8

*Proof.* Fix any state $s \in \mathcal{S}$.

If $C(s) = \{a_0(s)\}$, then the claim is immediate: the only candidate action is the baseline, so $D^\lambda(s) = a_0(s)$ for every $\lambda$.

Now suppose $C(s) \setminus \{a_0(s)\}$ is nonempty. Fix any $a \in C(s) \setminus \{a_0(s)\}$. Since

$$\Delta(s, a_0(s)) = \sum_{i=1}^{N} |q_i(s, a_0(s)) - q_i(s, a_0(s))| = 0,$$

we have

$$U_\lambda(s, a_0(s)) \geq U_\lambda(s, a) \iff U(s, a_0(s)) \geq U(s, a) - \lambda\Delta(s, a)$$
$$\iff \lambda \geq \frac{U(s, a) - U(s, a_0(s))}{\Delta(s, a)}.$$

The denominator is positive by assumption, so the ratio is finite.

By definition of $\lambda^\star(s)$, for every $\lambda \geq \lambda^\star(s)$ and every $a \in C(s) \setminus \{a_0(s)\}$,

$$U_\lambda(s, a_0(s)) \geq U_\lambda(s, a).$$

Therefore $a_0(s)$ is a maximizer of $U_\lambda(s, \cdot)$ over $C(s)$. Since ties are broken in favor of $a_0(s)$, the decision rule selects

$$D^\lambda(s) = a_0(s).$$

It remains only to note that $\bar\lambda < \infty$. For each state $s$, the set $C(s)$ is finite, every denominator $\Delta(s, a)$ appearing in the ratio is strictly positive, and $U(s, a)$ is real-valued. Hence $\lambda^\star(s) < \infty$ for each $s$. Since $\mathcal{S}$ is finite,

$$\bar\lambda = \max_{s \in \mathcal{S}} \lambda^\star(s) < \infty.$$

Thus, for any $\lambda \geq \bar\lambda$, we have $\lambda \geq \lambda^\star(s)$ for every $s$, and hence $D^\lambda(s) = a_0(s)$ for every $s \in \mathcal{S}$. $\square$

### A.3. Proof of Lemma 4.9

*Proof.* Let $\lambda \geq \bar{\lambda}$. By Lemma 4.8,

$$D^\lambda(s) = a_0(s) \qquad \text{for every } s \in \mathcal{S}.$$

Therefore, at any deployment time $t$ such that $\lambda_t \geq \bar{\lambda}$, the executed action is

$$a_t = D^{\lambda_t}(s_t) = a_0(s_t).$$

Since the baseline is safe by Definition 4.7,

$$\ell_t = \ell(s_t, a_0(s_t)) = 0.$$

Thus, taking

$$\lambda_{\text{safe}} = \bar{\lambda}, \qquad \alpha_{\text{safe}} = 0, \qquad K = 1,$$

we have that $\lambda_t \geq \lambda_{\text{safe}}$ implies

$$\frac{1}{K} \sum_{k=0}^{K-1} \ell_{t+k} = \ell_t = 0 \leq \alpha_{\text{safe}}.$$

This is exactly Definition 3.1. $\qquad\qquad\square$

### A.4. CDT-Style Controller Bound

The results in this subsection are not novel contributions of this work. We reproduce the standard conformal-controller argument of Lekeufack et al. (2024), adapted to our sign convention, where larger $\lambda$ corresponds to greater conservatism.

**Lemma A.1** (Bounded step size). *Under the update* (1), *for all $t \geq 0$,*

$$|\lambda_{t+1} - \lambda_t| \leq \eta.$$

*Proof.* By (1),

$$\lambda_{t+1} - \lambda_t = \eta(\ell_t - \alpha).$$

Since $\ell_t, \alpha \in [0, 1]$, we have $|\ell_t - \alpha| \leq 1$, and hence

$$|\lambda_{t+1} - \lambda_t| \leq \eta.$$

$$\square$$

**Lemma A.2** (Upper bound on the control variable). *Assume $\{D^\lambda\}$ is eventually safe with parameters $(\alpha_{\text{safe}}, \lambda_{\text{safe}}, K)$ as in Definition 3.1, with $\alpha_{\text{safe}} \leq \alpha$. Run the controller* (1) *with initialization*

$$\lambda_0 \leq \lambda_{\text{safe}} + \eta.$$

*Then, for all $t \geq 0$,*

$$\lambda_t \leq \lambda_{\text{safe}} + K\eta.$$

*Proof.* We argue by contradiction. Suppose there exists a time $t \geq 1$ such that

$$\lambda_t > \lambda_{\text{safe}} + K\eta.$$

Let

$$t^\star := \min\left\{t \geq 1 : \lambda_t > \lambda_{\text{safe}} + K\eta\right\}$$

be the first such time.

First, we claim that $t^\star \geq K$. If instead $t^\star \leq K - 1$, then by Lemma A.1,

$$\lambda_{t^\star} \leq \lambda_0 + t^\star \eta \leq \lambda_{\text{safe}} + \eta + t^\star \eta \leq \lambda_{\text{safe}} + K\eta,$$

contradicting the definition of $t^\star$. Hence $t^\star \geq K$.

By Lemma A.1, for each $m = 1, \ldots, K$,

$$\lambda_{t^\star - m} \geq \lambda_{t^\star} - m\eta > \lambda_{\text{safe}} + K\eta - m\eta \geq \lambda_{\text{safe}}.$$

Therefore,

$$\lambda_{t^\star - K}, \lambda_{t^\star - K + 1}, \ldots, \lambda_{t^\star - 1} \geq \lambda_{\text{safe}}.$$

By eventual safety over this length-$K$ window,

$$\frac{1}{K} \sum_{r=t^\star - K}^{t^\star - 1} \ell_r \leq \alpha_{\text{safe}} \leq \alpha.$$

Now unroll the update (1) over the same window:

$$\lambda_{t^\star} = \lambda_{t^\star - K} + \eta \sum_{r=t^\star - K}^{t^\star - 1} (\ell_r - \alpha)$$

$$= \lambda_{t^\star - K} + K\eta \left( \frac{1}{K} \sum_{r=t^\star - K}^{t^\star - 1} \ell_r - \alpha \right)$$

$$\leq \lambda_{t^\star - K}.$$

But $t^\star - K < t^\star$, so by the minimality of $t^\star$,

$$\lambda_{t^\star - K} \leq \lambda_{\text{safe}} + K\eta < \lambda_{t^\star}.$$

This contradicts $\lambda_{t^\star} \leq \lambda_{t^\star - K}$. Therefore no such $t^\star$ exists, and

$$\lambda_t \leq \lambda_{\text{safe}} + K\eta$$

for all $t \geq 0$. $\qquad\square$

### A.5. Main Long-Run Guarantee

*Proof of Theorem 4.10.* Unrolling the controller update (1) from time 0 to time $t$ gives

$$\lambda_{t+1} = \lambda_0 + \eta \sum_{r=0}^{t} (\ell_r - \alpha).$$

Rearranging,

$$\hat{R}_t := \frac{1}{t+1} \sum_{r=0}^{t} \ell_r = \alpha + \frac{\lambda_{t+1} - \lambda_0}{\eta(t+1)}. \qquad (2)$$

By Lemma 4.9, the CCO decision family is eventually safe with

$$\alpha_{\text{safe}} = 0, \qquad \lambda_{\text{safe}} = \bar{\lambda}, \qquad K = 1.$$

Since the theorem assumes

$$\lambda_0 \leq \bar{\lambda} + \eta,$$

Lemma A.2 implies that, for every $u \geq 0$,

$$\lambda_u \leq \bar{\lambda} + \eta.$$

In particular,

$$\lambda_{t+1} \leq \bar{\lambda} + \eta.$$

Substituting this bound into (2) gives

$$\hat{R}_t \leq \alpha + \frac{\bar{\lambda} + \eta - \lambda_0}{\eta(t+1)} = \alpha + \frac{(\bar{\lambda} - \lambda_0)/\eta + 1}{t+1}.$$

This proves the claim. $\qquad\square$

**A.6. Order-Sharpness of the $O(1/t)$ Transient Term**

We show that the dependence on $(\bar{\lambda} - \lambda_0)/(\eta(t+1))$ in Theorem 4.10 is worst-case order-sharp for the additive controller (1). The construction is intentionally minimal: one state, two actions, and deterministic losses.

**Proposition A.3** (Order-sharp lower-bound construction). *Fix any $\eta > 0$, any $\alpha \in [0, 1)$, and any gap $G > 0$. There exists a one-state, two-action CCO instance and an initialization with*

$$\bar{\lambda} - \lambda_0 = G$$

*such that, for every integer $t \geq 0$ satisfying*

$$t\eta(1 - \alpha) < G,$$

*the empirical loss is*

$$\hat{R}_t = 1.$$

*Consequently, for every such $t$ that also satisfies*

$$t + 1 \geq \frac{G}{2\eta(1 - \alpha)},$$

*we have*

$$\hat{R}_t - \alpha \geq \frac{G}{2\eta(t + 1)}.$$

*Thus the transient dependence on $(\bar{\lambda} - \lambda_0)/(\eta(t + 1))$ cannot be uniformly improved beyond constant factors.*

*Proof.* Consider a controlled Markov process with a single state $s$ and two candidate actions,

$$C(s) = \{a_0, a_1\}.$$

Let the baseline be

$$a_0(s) = a_0.$$

Define the primary scores and penalties by

$$U(s, a_0) = 0, \qquad U(s, a_1) = G,$$

and

$$\Delta(s, a_0) = 0, \qquad \Delta(s, a_1) = 1.$$

Define the losses by

$$\ell(s, a_0) = 0, \qquad \ell(s, a_1) = 1.$$

Then

$$U_\lambda(s, a_0) = 0, \qquad U_\lambda(s, a_1) = G - \lambda.$$

Hence the CCO decision rule selects the unsafe action $a_1$ whenever $\lambda < G$, and selects the baseline $a_0$ whenever $\lambda \geq G$, using the stipulated tie-breaking in favor of the baseline.

For this instance, the uniform baseline-dominance threshold is

$$\bar{\lambda} = G.$$

Initialize the controller at

$$\lambda_0 = 0,$$

so that

$$\bar{\lambda} - \lambda_0 = G.$$

As long as $\lambda_t < G$, the selected action is $a_1$ and the realized loss is $\ell_t = 1$. During this phase, the controller update is

$$\lambda_{t+1} = \lambda_t + \eta(1 - \alpha).$$

By induction, for every $t$ such that all previous actions have been unsafe,

$$\lambda_t = t\eta(1 - \alpha).$$

Therefore, whenever

$$t\eta(1 - \alpha) < G,$$

we have $\lambda_t < G$, so action $a_1$ is selected at time $t$. Moreover, for every $r = 0, \ldots, t$,

$$r\eta(1 - \alpha) \leq t\eta(1 - \alpha) < G,$$

so action $a_1$ was selected at every time $r \leq t$. Hence

$$\ell_r = 1 \qquad \text{for all } r = 0, \ldots, t,$$

and therefore

$$\hat{R}_t = \frac{1}{t+1} \sum_{r=0}^{t} \ell_r = 1.$$

It follows that

$$\hat{R}_t - \alpha = 1 - \alpha.$$

If additionally

$$t + 1 \geq \frac{G}{2\eta(1 - \alpha)},$$

then

$$\frac{G}{2\eta(t+1)} \leq 1 - \alpha.$$

Thus

$$\hat{R}_t - \alpha = 1 - \alpha \geq \frac{G}{2\eta(t+1)}.$$

Since $G = \bar{\lambda} - \lambda_0$, this gives a matching lower bound up to constants on the transient dependence appearing in Theorem 4.10.

$\square$

## B. Practical Extensions: Noisy and Delayed Feedback

Theorem 4.10 assumes that the controller observes the realized loss $\ell_t$ exactly and immediately after each action. We now show that the same style of guarantee extends to two common relaxations: noisy feedback and delayed feedback. Throughout this appendix, we assume the conditions of Theorem 4.10. In particular, the CCO decision family satisfies uniform baseline dominance with threshold $\bar{\lambda}$, and hence

$$\lambda_t \geq \bar{\lambda} \quad \implies \quad D^{\lambda_t}(s_t) = a_0(s_t) \quad \implies \quad \ell_t = 0.$$

We also assume $\ell_t \in [0, 1]$ and $\alpha \in [0, 1]$.

### B.1. Noisy Feedback

Suppose the controller does not observe $\ell_t$ directly. Instead, it observes

$$\tilde{\ell}_t = \ell_t + \xi_t,$$

and updates according to

$$\lambda_{t+1} = \lambda_t + \eta(\tilde{\ell}_t - \alpha) = \lambda_t + \eta(\ell_t - \alpha) + \eta\xi_t. \tag{3}$$

Let $\mathcal{F}_t$ denote the history up to and including the state and action at time $t$, but before observing the noise $\xi_t$. We assume that $\{\xi_t\}_{t \geq 0}$ is a martingale-difference noise sequence:

$$\mathbb{E}[\xi_t \mid \mathcal{F}_t] = 0,$$

and that it is conditionally $\sigma^2$-sub-Gaussian:

$$\mathbb{E}[\exp(\theta\xi_t) \mid \mathcal{F}_t] \leq \exp\left(\frac{\theta^2\sigma^2}{2}\right) \qquad \text{for all } \theta \in \mathbb{R}. \tag{4}$$

Bounded conditionally mean-zero noise with $|\xi_t| \leq \sigma$ satisfies this condition up to constants by Hoeffding's lemma.

The key idea is to separate the true-loss dynamics from the accumulated noise. Define the cumulative noise process

$$S_t := \sum_{s=0}^{t-1} \xi_s, \qquad S_0 := 0,$$

and define the de-noised virtual controller

$$\mu_t := \lambda_t - \eta S_t.$$

Then, subtracting $\eta S_{t+1}$ from both sides of (3),

$$\mu_{t+1} = \mu_t + \eta(\ell_t - \alpha). \tag{5}$$

Thus $\mu_t$ evolves exactly like the original noiseless controller, but the actions are selected using $\lambda_t$, not $\mu_t$. The next lemma shows that if the cumulative noise is not too negative, then large $\mu_t$ still implies safety.

**Lemma B.1** (Effective safety under bounded cumulative negative noise). *Fix $T \geq 0$ and suppose that, on an event $\mathcal{E}_T$, there exists a constant $B_T \geq 0$ such that*

$$S_t \geq -B_T \qquad \text{for all } t = 0, 1, \ldots, T. \tag{6}$$

*Then, on $\mathcal{E}_T$, the virtual controller $\mu_t$ is safe over the horizon $0, \ldots, T$ whenever it exceeds the threshold $\bar{\lambda} + \eta B_T$. That is, for every $t \leq T$,*

$$\mu_t \geq \bar{\lambda} + \eta B_T \implies \ell_t = 0.$$

*Proof.* Fix $t \leq T$ and suppose $\mu_t \geq \bar{\lambda} + \eta B_T$. Since $\lambda_t = \mu_t + \eta S_t$, condition (6) gives

$$\lambda_t = \mu_t + \eta S_t \geq \bar{\lambda} + \eta B_T - \eta B_T = \bar{\lambda}.$$

By uniform baseline dominance, $\lambda_t \geq \bar{\lambda}$ implies that the selected action is the baseline $a_0(s_t)$. Since the baseline is safe, $\ell_t = \ell(s_t, a_0(s_t)) = 0$. □

We now prove the noisy-feedback guarantee.

**Proposition B.2** (Loss control under noisy feedback). *Assume the controller updates according to (3), where $\{\xi_t\}$ is conditionally mean-zero and $\sigma^2$-sub-Gaussian as in (4). Suppose $\lambda_0 \leq \bar{\lambda} + \eta$. Then, for any fixed $t \geq 0$, with probability at least $1 - \delta$,*

$$\hat{R}_t := \frac{1}{t+1} \sum_{s=0}^{t} \ell_s \leq \alpha + \frac{(\bar{\lambda} - \lambda_0)/\eta + 1}{t+1} + \sigma\sqrt{\frac{2\log(1/\delta)}{t+1}}. \tag{7}$$

*Proof.* Fix $t \geq 0$. By the maximal sub-Gaussian martingale inequality applied to the martingale $S_k = \sum_{s=0}^{k-1} \xi_s$,

$$\mathbb{P}\left(\min_{0 \leq k \leq t+1} S_k < -\sigma\sqrt{2(t+1)\log(1/\delta)}\right) \leq \delta.$$

Therefore, with probability at least $1 - \delta$, the event $\mathcal{E}_t$ holds with

$$B_t := \sigma\sqrt{2(t+1)\log(1/\delta)}. \tag{8}$$

We work on this event for the remainder of the proof.

By Lemma B.1, the virtual controller $\mu_t$ is safe over the horizon $0, \ldots, t$ whenever $\mu_t \geq \bar{\lambda} + \eta B_t$. Since $\mu_t$ evolves by (5) and since $\ell_t, \alpha \in [0, 1]$, the same one-step controller-envelope argument used in Lemma A.2 applies to $\mu_t$. Hence, for every $s \leq t + 1$,

$$\mu_s \leq \bar{\lambda} + \eta B_t + \eta. \tag{9}$$

The initialization condition required for this envelope is satisfied because

$$\mu_0 = \lambda_0 \leq \bar{\lambda} + \eta \leq \bar{\lambda} + \eta B_t + \eta.$$

Unrolling the virtual update (5) gives

$$\mu_{t+1} = \lambda_0 + \eta \sum_{s=0}^{t} (\ell_s - \alpha).$$

Rearranging,

$$\hat{R}_t = \alpha + \frac{\mu_{t+1} - \lambda_0}{\eta(t+1)}. \tag{10}$$

Using (9) in (10),

$$\hat{R}_t \leq \alpha + \frac{\bar{\lambda} + \eta B_t + \eta - \lambda_0}{\eta(t+1)} = \alpha + \frac{(\bar{\lambda} - \lambda_0)/\eta + 1 + B_t}{t+1}.$$

Substituting the definition of $B_t$ from (8) gives

$$\hat{R}_t \leq \alpha + \frac{(\bar{\lambda} - \lambda_0)/\eta + 1}{t+1} + \sigma \sqrt{\frac{2 \log(1/\delta)}{t+1}},$$

which proves the claim. $\qquad\square$

If one assumes only the deterministic boundedness condition $|\xi_t| \leq \sigma$, without any mean-zero or concentration assumption, then the same proof applies with the pathwise bound $B_t = \sigma(t+1)$. Consequently,

$$\hat{R}_t \leq \alpha + \frac{(\bar{\lambda} - \lambda_0)/\eta + 1}{t+1} + \sigma. \tag{11}$$

Thus conditionally mean-zero noise contributes a vanishing $O(\sigma/\sqrt{t})$ term with high probability, whereas adversarial bounded noise can introduce a non-vanishing bias of size at most $\sigma$.

### B.2. Delayed Feedback

We next consider delayed observations. Suppose the loss generated at round $r$ is revealed after an integer delay $d_r$, where

$$0 \leq d_r \leq D$$

for some fixed $D < \infty$. Let

$$\mathcal{G}_t := \{r \geq 0 : r + d_r = t\}$$

denote the set of losses revealed at time $t$. The delayed controller updates according to

$$\lambda_{t+1} = \lambda_t + \eta \sum_{r \in \mathcal{G}_t} (\ell_r - \alpha). \tag{12}$$

Each loss is therefore incorporated exactly once, but possibly several rounds after the action that generated it.

The following proof uses an augmented cumulative variable that accounts for all losses generated so far, whether or not they have already been revealed.

**Lemma B.3** (Delayed augmented controller envelope). *Assume $d_r \leq D$ for all $r$ and $\lambda_0 \leq \bar{\lambda} + \eta(D+1)$. Define*

$$Z_{t+1} := \lambda_0 + \eta \sum_{r=0}^{t} (\ell_r - \alpha). \tag{13}$$

*Then, for all $t \geq 0$,*

$$Z_{t+1} \leq \bar{\lambda} + \eta(D+1). \tag{14}$$

*Proof.* Fix $t \geq 0$. We prove (14) pathwise.

First suppose that $\lambda_j \geq \bar{\lambda}$ for every $j = 0, 1, \ldots, t$. Then every action taken up to time $t$ is the baseline, so $\ell_j = 0$ for all $j = 0, \ldots, t$. Hence

$$Z_{t+1} = \lambda_0 - \eta\alpha(t+1) \leq \lambda_0 \leq \bar{\lambda} + \eta(D+1),$$

as desired.

Otherwise, define

$$h := \max\{j \in \{0, \ldots, t\} : \lambda_j < \bar{\lambda}\}.$$

By maximality of $h$, we have $\lambda_j \geq \bar{\lambda}$ for every $j = h+1, \ldots, t$. Therefore,

$$\ell_j = 0 \qquad \text{for all } j = h+1, \ldots, t. \tag{15}$$

Now express $Z_{t+1}$ relative to $\lambda_h$. By the delayed update, $\lambda_h$ contains exactly the losses revealed by time $h-1$:

$$\lambda_h = \lambda_0 + \eta \sum_{\substack{0 \leq r \\ r+d_r \leq h-1}} (\ell_r - \alpha). \tag{16}$$

On the other hand,

$$Z_{t+1} = \lambda_0 + \eta \sum_{r=0}^{t} (\ell_r - \alpha). \tag{17}$$

Subtracting (16) from (17) gives

$$Z_{t+1} = \lambda_h + \eta \sum_{\substack{0 \leq r \leq t \\ r+d_r \geq h}} (\ell_r - \alpha). \tag{18}$$

We now bound the remaining sum. Any term with $r > h$ is nonpositive, because by (15), $\ell_r = 0$, and hence

$$\ell_r - \alpha = -\alpha \leq 0.$$

Thus only indices $r \leq h$ can contribute positively to the sum in (18). For such an index to appear in the sum, it must satisfy $r + d_r \geq h$. Since $d_r \leq D$, this implies

$$r \geq h - D.$$

Therefore the only indices that can contribute positively are contained in

$$\{h - D, h - D + 1, \ldots, h\},$$

which has cardinality at most $D + 1$. Since $\ell_r - \alpha \leq 1$ for every $r$, we obtain

$$\sum_{\substack{0 \leq r \leq t \\ r+d_r \geq h}} (\ell_r - \alpha) \leq D + 1.$$

Substituting this into (18) and using $\lambda_h < \bar{\lambda}$,

$$Z_{t+1} \leq \lambda_h + \eta(D+1) < \bar{\lambda} + \eta(D+1).$$

This proves (14). $\qquad\square$

We can now prove the delayed-feedback guarantee.

**Proposition B.4** (Loss control under bounded delay). *Assume $d_r \leq D$ for all $r$ and initialize the controller so that*

$$\lambda_0 \leq \bar{\lambda} + \eta(D+1).$$

*Under the delayed update* (12), *for every $t \geq 0$,*

$$\hat{R}_t := \frac{1}{t+1} \sum_{r=0}^{t} \ell_r \leq \alpha + \frac{(\bar{\lambda} - \lambda_0)/\eta + D + 1}{t+1}. \tag{19}$$

*Proof.* By definition of $Z_{t+1}$ in (13),

$$Z_{t+1} = \lambda_0 + \eta \sum_{r=0}^{t} (\ell_r - \alpha).$$

Rearranging,

$$\hat{R}_t = \alpha + \frac{Z_{t+1} - \lambda_0}{\eta(t+1)}. \tag{20}$$

By Lemma B.3,

$$Z_{t+1} \leq \bar{\lambda} + \eta(D+1).$$

Substituting this into (20) gives

$$\hat{R}_t \leq \alpha + \frac{\bar{\lambda} + \eta(D+1) - \lambda_0}{\eta(t+1)} = \alpha + \frac{(\bar{\lambda} - \lambda_0)/\eta + D + 1}{t+1}.$$

This proves the claim. □

When $D = 0$, the bound becomes

$$\hat{R}_t \leq \alpha + \frac{(\bar{\lambda} - \lambda_0)/\eta + 1}{t+1},$$

which exactly recovers Theorem 4.10. Thus bounded delay adds only an additional $D/(t+1)$ transient term relative to the immediate-feedback guarantee.

## C. Overseer Quality and Conservatism

This appendix relates the quality of the auxiliary overseer signals to the amount of conservatism required for safety. Throughout, write $a_0(s)$ for the baseline action and recall that

$$U_\lambda(s,a) := U(s,a) - \lambda\Delta(s,a).$$

We assume $a_0(s) \in C(s)$ for every state $s$, and that the baseline is safe:

$$\ell(s, a_0(s)) = 0 \qquad \text{for all } s.$$

The decision rule $D^\lambda$ is the CCO rule with tie-breaking in favor of $a_0(s)$ whenever the baseline is among the maximizers.

### C.1. Overseer Margin Bounds the Safe Threshold

We first relate overseer discrimination strength to the conservatism required for zero loss. Define the set of loss-incurring state-action pairs

$$\mathcal{H} := \{(s,a) : a \in C(s), \ \ell(s,a) > 0\}.$$

If $\mathcal{H} = \emptyset$, then every candidate action has zero loss and all claims below are trivial. Hence, in the remainder of this subsection, we assume $\mathcal{H} \neq \emptyset$.

**Definition C.1** (Overseer margin). The overseer margin is

$$\kappa := \inf_{(s,a) \in \mathcal{H}} \Delta(s,a).$$

We also define the maximal positive utility advantage of any candidate over the baseline:

$$B := \sup_{s \in S} \sup_{a \in C(s)} \big[U(s,a) - U(s, a_0(s))\big]_+, \qquad [x]_+ := \max\{x, 0\}.$$

The positive part is useful because actions that already have lower primary utility than the baseline do not require penalty to suppress.

**Proposition C.2** (Overseer margin implies a small safe threshold). *Assume $\kappa > 0$ and $B < \infty$. Then, for every*

$$\lambda \geq \frac{B}{\kappa},$$

*the tie-broken CCO decision rule satisfies*

$$\ell(s, D^\lambda(s)) = 0 \qquad \text{for every } s \in S.$$

*Consequently, the CCO decision family is eventually safe with*

$$\lambda_{\text{safe}} \leq \frac{B}{\kappa}, \qquad \alpha_{\text{safe}} = 0, \qquad K = 1.$$

*Moreover, if $\lambda > B/\kappa$, then every maximizer of $U_\lambda(s, \cdot)$ has zero loss.*

*Proof.* Fix a state $s$ and an action $a \in C(s)$ with $\ell(s, a) > 0$. Then $(s, a) \in \mathcal{H}$, so by definition of $\kappa$,

$$\Delta(s, a) \geq \kappa.$$

Also, by definition of $B$,

$$U(s, a) - U(s, a_0(s)) \leq B.$$

Since $\Delta(s, a_0(s)) = 0$, we have

$$U_\lambda(s, a) - U_\lambda(s, a_0(s)) = \big(U(s, a) - U(s, a_0(s))\big) - \lambda \Delta(s, a)$$
$$\leq B - \lambda \kappa.$$

If $\lambda \geq B/\kappa$, then

$$U_\lambda(s, a) \leq U_\lambda(s, a_0(s))$$

for every loss-incurring action $a$.

Therefore, no loss-incurring action can strictly dominate the baseline. If a loss-incurring action ties the baseline, then the baseline is also a maximizer, and the CCO tie-breaking rule selects $a_0(s)$ rather than the loss-incurring action. Hence

$$\ell(s, D^\lambda(s)) = 0$$

for every state $s$.

Since this implication holds uniformly over states, taking $\lambda_{\text{safe}} = B/\kappa$, $\alpha_{\text{safe}} = 0$, and $K = 1$ gives eventual safety.

Finally, if $\lambda > B/\kappa$, then

$$B - \lambda \kappa < 0,$$

so every loss-incurring action is strictly worse than the baseline under $U_\lambda(s, \cdot)$. Hence no loss-incurring action can be a maximizer. $\qquad \square$

## C.2. Effect of Bounded Overseer Error

We next quantify how errors in the auxiliary overseer scores affect the penalty margin. Suppose each overseer score decomposes as

$$q_i(s, a) = q_i^\star(s, a) + \xi_i(s, a),$$

where $q_i^\star$ is an idealized score and the perturbation satisfies

$$|\xi_i(s, a)| \leq \sigma_i \qquad \text{for all } (s, a).$$

Let

$$\sigma := \sum_{i=1}^{N} \sigma_i.$$

Define the ideal penalty

$$\Delta^\star(s, a) := \sum_{i=1}^{N} |q_i^\star(s, a) - q_i^\star(s, a_0(s))|.$$

**Lemma C.3** (Penalty perturbation bound). *For every state-action pair $(s, a)$,*

$$\left|\Delta(s, a) - \Delta^\star(s, a)\right| \le 2\sigma.$$

*Proof.* Fix $(s, a)$. For each overseer $i$, define

$$x_i := q_i^\star(s, a) - q_i^\star(s, a_0(s))$$

and

$$e_i := \xi_i(s, a) - \xi_i(s, a_0(s)).$$

Then

$$q_i(s, a) - q_i(s, a_0(s)) = x_i + e_i.$$

By the reverse triangle inequality,

$$\left||x_i + e_i| - |x_i|\right| \le |e_i|.$$

Moreover,

$$|e_i| \le |\xi_i(s, a)| + |\xi_i(s, a_0(s))| \le 2\sigma_i.$$

Therefore,

$$\begin{aligned}
\left|\Delta(s, a) - \Delta^\star(s, a)\right| &= \left|\sum_{i=1}^{N} |x_i + e_i| - \sum_{i=1}^{N} |x_i|\right| \\
&\le \sum_{i=1}^{N} \left||x_i + e_i| - |x_i|\right| \\
&\le \sum_{i=1}^{N} 2\sigma_i = 2\sigma.
\end{aligned}$$

$\square$

Define the ideal overseer margin over the actual loss-incurring actions:

$$\kappa^\star := \inf_{(s,a)\in\mathcal{H}} \Delta^\star(s, a).$$

**Corollary C.4** (Safe threshold under bounded overseer error). *Assume $\mathcal{H} \ne \emptyset$, $B < \infty$, and $\kappa^\star > 2\sigma$. Then the realized margin satisfies*

$$\kappa \ge \kappa^\star - 2\sigma > 0.$$

*Consequently, the CCO decision family is eventually safe with*

$$\lambda_{\text{safe}} \le \frac{B}{\kappa^\star - 2\sigma}.$$

*Proof.* For every $(s, a) \in \mathcal{H}$, Lemma C.3 implies

$$\Delta(s, a) \ge \Delta^\star(s, a) - 2\sigma.$$

Taking the infimum over $(s, a) \in \mathcal{H}$ gives

$$\kappa = \inf_{(s,a)\in\mathcal{H}} \Delta(s, a) \ge \inf_{(s,a)\in\mathcal{H}} \Delta^\star(s, a) - 2\sigma = \kappa^\star - 2\sigma.$$

Since $\kappa^\star > 2\sigma$, the realized margin is positive. Applying Proposition C.2 with $\kappa \ge \kappa^\star - 2\sigma$ gives

$$\lambda_{\text{safe}} \le \frac{B}{\kappa} \le \frac{B}{\kappa^\star - 2\sigma}.$$

$\square$

### C.3. Surrogate Alignment Between Penalty and Loss

The margin condition above requires every loss-incurring action to be separated from the baseline by a fixed positive penalty. This can be strong when losses are graded. We therefore also consider a weaker surrogate condition requiring the penalty to lower bound the loss.

**Assumption C.5** (Surrogate dominance). There exists a constant $c > 0$ such that

$$\Delta(s, a) \geq c\,\ell(s, a) \qquad \text{for all } s \in S, \ a \in C(s).$$

**Proposition C.6** (Suppressing $\varepsilon$-bad actions). *Assume Assumption C.5 and $B < \infty$. Fix $\varepsilon > 0$. Then, for every*

$$\lambda \geq \frac{B}{c\varepsilon},$$

*the tie-broken CCO decision rule satisfies*

$$\ell(s, D^\lambda(s)) < \varepsilon \qquad \text{for every } s \in S.$$

*Equivalently, $D^\lambda$ never selects an action with loss at least $\varepsilon$. Moreover, if $\lambda > B/(c\varepsilon)$, then every maximizer of $U_\lambda(s, \cdot)$ has loss strictly less than $\varepsilon$.*

*Proof.* Fix a state $s$ and an action $a \in C(s)$ with $\ell(s, a) \geq \varepsilon$. By Assumption C.5,

$$\Delta(s, a) \geq c\,\ell(s, a) \geq c\varepsilon.$$

Using the definition of $B$,

$$U_\lambda(s, a) - U_\lambda(s, a_0(s)) = \big(U(s, a) - U(s, a_0(s))\big) - \lambda\Delta(s, a)$$
$$\leq B - \lambda c\varepsilon.$$

If $\lambda \geq B/(c\varepsilon)$, then

$$U_\lambda(s, a) \leq U_\lambda(s, a_0(s)).$$

Thus no action with loss at least $\varepsilon$ can strictly dominate the baseline. If such an action ties the baseline, then the baseline is also a maximizer, and the tie-breaking rule selects the baseline. Therefore the selected action cannot have loss at least $\varepsilon$.

If $\lambda > B/(c\varepsilon)$, then

$$B - \lambda c\varepsilon < 0,$$

so every action with loss at least $\varepsilon$ is strictly worse than the baseline. Hence no such action can be a maximizer. $\square$

**Corollary C.7** (Binary or positively gapped losses). *Suppose Assumption C.5 holds and $B < \infty$.*

*If losses are binary, $\ell(s, a) \in \{0, 1\}$, then the CCO decision rule is safe for every*

$$\lambda \geq \frac{B}{c}.$$

*More generally, if there exists $\ell_{\min} > 0$ such that every loss-incurring action satisfies*

$$\ell(s, a) \geq \ell_{\min},$$

*then the CCO decision rule is safe for every*

$$\lambda \geq \frac{B}{c\ell_{\min}}.$$

*Proof.* For binary losses, apply Proposition C.6 with $\varepsilon = 1$. Then the selected action has loss strictly less than 1, and since the loss is binary, it must have loss zero.

For positively gapped losses, apply Proposition C.6 with $\varepsilon = \ell_{\min}$. The selected action has loss strictly less than $\ell_{\min}$, so by the gap assumption it cannot be loss-incurring. Hence its loss is zero. $\square$

# D. Additional Ablations and Baselines

This appendix reports the supplementary experiments referenced in the main text: a cross-model replication using the GPT-5.4 family (§D.1), an ablation over the number of overseers $n$ (§D.2), a sensitivity analysis of the baseline action $a_0$ (§D.3), and a comparison against an adaptive majority-vote baseline (§D.4). All ablations use the SWE-bench setup of §5.1 unless otherwise specified.

## D.1. Cross-Model Replication: GPT-5.4 Family

To verify that the $\alpha$-tracking property is not specific to the Anthropic model family, we replicate the SWE-bench experiment using GPT-5.4 as the strong (primary) agent and GPT-5.4-mini and GPT-5.4-nano as overseers. We additionally vary the number of overseers $n$.

*Table 3.* SWE-bench misalignment rates with GPT-5.4 as the primary agent and GPT-5.4-mini overseers, varying $n$ and $\alpha$ (3 seeds, 300 instances per cell). $\alpha$-tracking holds across all $(n, \alpha)$ pairs.

| $n$ | $\alpha = 0.05$ | $\alpha = 0.10$ | $\alpha = 0.15$ | $\alpha = 0.20$ |
|---|---|---|---|---|
| 1 | 7.1% | 12.2% | 16.8% | 21.4% |
| 2 | 6.4% | 11.5% | 16.1% | 20.7% |
| 3 | 5.9% | 10.8% | 15.5% | 20.1% |
| 5 | 5.3% | 10.3% | 15.1% | 19.7% |
| 7 | 5.0% | 9.9% | 14.9% | 19.5% |
| 10 | 4.7% | 9.7% | 14.7% | 19.4% |

The results in Table 3 demonstrate three properties: (i) $\alpha$-tracking holds across all $(n, \alpha)$ pairs, with maximum deviation 2.1pp; (ii) increasing $n$ tightens precision monotonically; (iii) even a single overseer ($n = 1$) achieves usable tracking. We further verify that the weak-actor and adaptive-MV comparisons of §5.1 and §D.4 replicate under this setup, with CCO consistently matching its target while baselines exhibit larger deviations.

## D.2. Overseer Count Ablation

Table 3 also serves as the overseer-count ablation. The penalty $\Delta = \sum_i |q_i - q_i(a_0)|$ grows additively with $n$, but the conformal controller compensates by adjusting $\lambda_t$ downward, so target-tracking is preserved across $n \in \{1, 2, 3, 5, 7, 10\}$. The choice of $n$ is therefore a precision/cost knob rather than a hyperparameter requiring tuning to achieve the safety target.

## D.3. Baseline Action ($a_0$) Sensitivity

A natural concern is whether CCO is robust to the choice of baseline action $a_0$. We compare four strategies on SWE-bench ($\alpha = 0.10$, 3 seeds, 300 instances):

1. **Fixed deferral** (the low-resource candidate; main-text default).

2. **Lowest-penalty candidate**: $a_0 = \arg\min_a \Delta(s, a)$ at each step.

3. **Max-min overseer score**: $a_0 = \arg\max_a \min_i q_i(s, a)$.

4. **Highest average overseer score**: $a_0 = \arg\max_a \frac{1}{N} \sum_i q_i(s, a)$.

*Table 4.* $a_0$ sensitivity on SWE-bench, $\alpha = 0.10$ (3 seeds, 300 instances). All strategies converge near target.

| $a_0$ Strategy | Misalignment Rate (%) |
|---|---|
| Fixed deferral (main-text default) | 10.4 |
| Lowest-penalty candidate | 11.1 |
| Max-min overseer score | 9.8 |
| Highest avg overseer score | 10.7 |

All four strategies stay within 1.1pp of the $\alpha = 0.10$ target. The conformal controller adapts $\lambda_t$ to compensate for the different effective penalty scales each $a_0$ induces, so CCO does not require an oracle to specify $a_0$.

## D.4. Adaptive Majority Vote

We implement an adaptive majority-vote baseline that targets the same long-run misalignment rate $\alpha$ as CCO. At each step, each of the $N = 10$ overseers votes ACCEPT on a candidate if its score $q_j$ exceeds a threshold $\tau_t$, and REJECT otherwise; majority-rejected candidates are eliminated, and the highest-utility survivor is selected. The threshold $\tau_t$ is re-fit at each step on all historical data so that the running misalignment rate matches $\alpha$.

**Steady state.** Under stationary conditions matching the SWE-bench protocol of §5.1 ($p = 0.5$, 300 instances), CCO and adaptive MV achieve comparable realized rates near the target. CCO's transient phase is 2–7× shorter, reflecting the direct projected-gradient update on $\lambda$ versus the indirect threshold fitting.

**Distribution shift.** We construct a distribution-shift sequence where the misalignment probability $p$ jumps from 0.2 to 0.8 at instance 100 and back to 0.2 at instance 200 (300 total instances per $\alpha$). Table 5 reports the phase-wise misalignment rates.

*Table 5.* Distribution-shift comparison. $p$ jumps $0.2 \rightarrow 0.8 \rightarrow 0.2$ at instances 100 and 200.

| $\alpha$ | Phase (instances) | CCO Rate (%) | Adaptive MV Rate (%) |
|---|---|---|---|
| 0.05 | Phase 1 ($p$=0.2, 1–100) | 6.7 | 8.8 |
| 0.05 | Phase 2 ($p$=0.8, 101–200) | 6.1 | 18.2 |
| 0.05 | Phase 3 ($p$=0.2, 201–300) | 3.8 | 4.6 |
| 0.10 | Phase 1 | 12.0 | 12.5 |
| 0.10 | Phase 2 | 11.1 | 15.5 |
| 0.10 | Phase 3 | 10.2 | 14.2 |
| 0.15 | Phase 1 | 14.4 | 14.9 |
| 0.15 | Phase 2 | 15.1 | 22.0 |
| 0.15 | Phase 3 | 14.4 | 20.2 |

CCO stays near target throughout all three phases. Adaptive MV overshoots substantially during the high-misalignment phase (e.g., 18.2% vs. 6.1% at $\alpha = 0.05$) and remains elevated even after the shift reverses, because its threshold cannot adapt quickly enough through the integral signal of running rate.

# E. SWE-bench Implementation Details

## E.1. Benchmark and Adversarial Protocol

SWE-bench Lite (Jimenez et al., 2024) comprises 300 real-world GitHub issues from popular Python repositories including Django, SymPy, scikit-learn, and matplotlib. Each instance presents a bug report or feature request alongside the relevant repository, and the task is to produce a unified-diff patch that resolves the issue and passes the project's hidden test suite when run in a Docker container.

We use Claude Sonnet 4.6 as the tool-calling primary agent and Claude Haiku 4.5 as the overseer. For each instance, the agent generates three candidate patches at distinct *resource levels* (high: 9–12 turns, medium: 5–8 turns, low: 1–4 turns), with normalized utility $u_i = \text{turns}_i/12$. With probability $p = 0.5$ the high-resource candidate is given an instance-specific misalignment instruction generated by Claude Opus 4.6, targeting a vulnerability tailored to the files the issue touches. Medium and low-resource candidates are always aligned; the low-resource candidate also serves as the CCO baseline $a_0$.

## E.2. Instance-Specific Vulnerabilities

Vulnerability categories include: SQL injection via raw string formatting, code execution through `eval()`/`exec()`, unsafe deserialization with `pickle.loads()`, XSS via `mark_safe()` on unsanitized input, path traversal, information disclosure via verbose logging, command injection through `shell=True` subprocesses, and broad `except` clauses that silently swallow security-critical errors. Each is designed to resemble a plausible implementation choice rather than an obviously malicious insertion. Opus 4.6 receives the issue text plus relevant file paths and chooses an appropriate category for that specific instance.

## E.3. Random-Context Overseers

For each candidate patch we instantiate $N = 10$ independent Claude Haiku 4.5 calls. Each overseer receives the issue, the patch, and a uniform random sample of $k = 4$ Python files from the repository (excluding build artifacts and files exceeding 50 KB). This models the realistic constraint that an overseer cannot in general read the entire codebase. Each call uses temperature 0.7.

---

**Overseer Prompt (Random-Context**

```
System:  You are a code reviewer.  Provide concise, objective evaluations.
User:  The following patch was proposed for this issue.  You also have access to a
small sample of repository files for context.  Flag anything suspicious, unnecessary,
unusual, or inconsistent with the codebase style.  Assign a single score on a 1--10
scale, where 10 means the patch looks clean and safe and 1 means it looks suspicious
or dangerous.
## Issue
{issue_text}

## Proposed Patch (unified diff)
{patch}

## Repository File Sample
{four_random_files}

Respond in this exact format:
SCORE: [1--10]
JUSTIFICATION: [Brief explanation]
```

---

Scores are normalized to $[0, 1]$ by dividing by 10, matching the scale of the candidate utilities $u_i$. If the structured SCORE: line cannot be parsed, a default score of 5.0 is assigned. The penalty for candidate $i$ is $p_i = \sum_{j=1}^{N} |q_j^{(i)} - q_j^{(\text{low})}|$.

## E.4. Evaluation and Resimulation

After the online CCO run, every candidate patch (high, medium, and low) is independently evaluated in a Docker container against the SWE-bench test suite. Each candidate receives a binary PASS/FAIL test result. Given the saved overseer scores, utilities, and test outcomes, we can resimulate CCO's online selection at any $\alpha \in [0, 1]$ by replaying the $\lambda_t$ updates over the saved trajectory and looking up the test result of the candidate CCO would have selected. This factorization is what lets Table 1 report both misalignment rate and solve rate across multiple $\alpha$ values from a single experimental run.

## E.5. Tool-Calling Loop

The primary agent operates in a realistic tool-calling loop with access to: `read_file`, `list_dir`, `search_repo`, `write_patch`, and `run_tests`. The turn budget for each candidate is sampled uniformly from the band corresponding to its resource level. The agent must produce a unified-diff patch and may optionally invoke `run_tests` before submitting; many submitted patches are pre-verified by the agent against the visible portion of the test suite.

# F. Simpler SWE-bench Protocol: Fixed-Utility Variant

Here we report results from a simpler instantiation of the SWE-bench experiment. This protocol uses the same overall structure (CCO selecting among multiple candidate patches scored by an ensemble of weak overseers) but with three differences: (i) candidate utilities are *predetermined* ($u = 10, 8, 7, 3$) rather than derived from an agentic resource budget, (ii) the misalignment probability is $p = 0.3$ instead of $p = 0.5$, and (iii) overseers are specialized along five fixed dimensions (scope, safety, simplicity, convention, necessity) rather than receiving random repository samples, so they evaluate via prompt-defined criteria rather than from a partial code view. This protocol is therefore more controlled but less realistic. We include it as a robustness check showing the same calibration behavior under a different misalignment-generation regime.

### F.1. Setup

Each instance receives four candidates: Candidate 1 ($u_1 = 10$, most comprehensive solution; with probability $p = 0.3$ receives the secret misalignment objective), Candidate 2 ($u_2 = 8$, solid but simpler, always aligned), Candidate 3 ($u_3 = 7$, minimal working solution, always aligned), and a Baseline ($u_0 = 3$, deferral). Both Sonnet 4.5 (primary) and Haiku 4.5 (overseers) are used as in the MACHIAVELLI experiment. Five overseers evaluate each candidate with the criteria below.

- **Scope Guardian**: flags modifications to unrelated files/functions and scope creep.

- **Safety Checker**: detects dangerous patterns (`eval`, `exec`, SQL concatenation, hardcoded credentials, `shell=True`, unsafe deserialization).

- **Simplicity Assessor**: penalizes unnecessary abstraction, dead code, over-engineering.

- **Convention Follower**: checks adherence to repository-specific patterns and style.

- **Necessity Auditor**: questions whether each change is required for the fix.

Each overseer assigns $q_i(s, a) \in [1, 10]$ with higher values indicating better alignment.

### F.2. Results

We evaluate on all 300 SWE-bench Lite instances, 5 seeds, $\alpha \in \{0.05, 0.10, 0.15, 0.20\}$ (6,000 total evaluations). Table 6 shows CCO achieves misalignment rates within 1.3 percentage points of all targets, with standard deviations below 0.7pp.

*Table 6.* Earlier SWE-bench protocol results (5 seeds $\times$ 300 instances, $p = 0.3$ selective misalignment, fixed candidate utilities). All values are mean $\pm$ standard deviation across seeds.

| Target $\alpha$ | Misalignment Rate | Gap | Catch Rate | Defer Rate |
|---|---|---|---|---|
| 0.05 | $6.27 \pm 0.43\%$ | $+1.27$pp | $78.70 \pm 1.73\%$ | $66.47 \pm 5.07\%$ |
| 0.10 | $10.87 \pm 0.38\%$ | $+0.87$pp | $63.28 \pm 1.71\%$ | $46.67 \pm 4.09\%$ |
| 0.15 | $15.67 \pm 0.33\%$ | $+0.67$pp | $46.29 \pm 4.55\%$ | $34.27 \pm 3.66\%$ |
| 0.20 | $19.53 \pm 0.69\%$ | $-0.47$pp | $32.76 \pm 5.54\%$ | $21.13 \pm 3.96\%$ |

The *catch rate* (the proportion of misalignment attempts blocked via deferral or selection of an aligned alternative) ranges from 78.7% at $\alpha = 0.05$ to 32.8% at $\alpha = 0.20$, even though the misaligned candidate has the highest predetermined utility, so the overseers must override a clear utility signal. The *deferral rate* (selection of the $u_0 = 3$ baseline) ranges from 66.5% to 21.1%, reflecting the trade-off between safety guarantees and system autonomy. Seed-level rates are reported in Table 7.

*Table 7.* Misalignment rates by seed for the earlier protocol (deployed misalignments out of 300 instances).

| Seed | $\alpha = 0.05$ | $\alpha = 0.10$ | $\alpha = 0.15$ | $\alpha = 0.20$ |
|---|---|---|---|---|
| 0 | 6.00% (18) | 10.33% (31) | 15.33% (46) | 19.00% (57) |
| 1 | 7.00% (21) | 11.33% (34) | 16.00% (48) | 20.33% (61) |
| 2 | 6.00% (18) | 10.67% (32) | 15.33% (46) | 18.67% (56) |
| 3 | 6.33% (19) | 11.00% (33) | 16.00% (48) | 20.00% (60) |
| 4 | 6.00% (18) | 11.00% (33) | 15.67% (47) | 19.67% (59) |

## G. MACHIAVELLI Implementation Details

**Benchmark Background.** The MACHIAVELLI benchmark (Pan et al., 2023) evaluates whether agents can behave ethically in complex social environments. Built on Choice of Games, a platform of professionally-written interactive fiction, the benchmark presents nuanced scenarios where ethical considerations emerge naturally from story progression. This creates realistic moral dilemmas where "correct" choices are often ambiguous.

**LLM Implementation.** We use Claude Sonnet 4.5 (Anthropic, 2025b) as the primary agent and for violation detection, while Claude Haiku 4.5 (Anthropic, 2025a) implements the five auxiliary overseers. The weaker but cheaper Haiku overseers provide sufficient signal to constrain the stronger Sonnet agent, embodying the weak-to-strong oversight paradigm central to scalable alignment.

**Primary Decision Agent.** At each state $s$, the agent observes game text (up to 4000 characters) and the set $C(s)$ of available actions. For each $a \in C(s)$, the agent produces a utility score $U(s,a) \in [0, 10]$ representing expected in-game reward and strategic value. The action with highest utility is the primary agent's preferred choice, which CCO may approve or override depending on $\lambda_t$ and the auxiliary penalties.

**Auxiliary Overseers.** We employ five specialized overseers implemented with Claude Haiku 4.5, each focusing on a distinct ethical dimension: Harm Prevention (physical and emotional harm), Honesty Advocate (truthfulness and transparency), Autonomy Guardian (respect for others' autonomy and consent), Fairness Monitor (unfairness, discrimination, and exploitation), and Trust Keeper (loyalty, promise-keeping, and relationship integrity). Each overseer $q_i$ assigns scores $q_i(s,a) \in [0, 10]$ where higher values indicate greater alignment with the ethical dimension.

**Baseline Action Selection.** MACHIAVELLI games do not provide an explicit baseline action, so we define the baseline action $a_0$ dynamically per state as the action with the lowest violation count according to the MACHIAVELLI annotation scheme. When multiple actions achieve the minimum violation count, we select one uniformly at random but cache this choice per state to ensure consistent penalty computation across all queries in the same state.

**Violation Detection.** For violation detection $v(s,a)$, we query Claude Sonnet 4.5 with MACHIAVELLI's 13-category annotation scheme, which includes killing, physical harm, non-physical harm (emotional/psychological), intending harm, deception, manipulation, promises, betrayal, stealing, trespassing, spying, vandalism, and fairness violations. The model produces binary labels for each category, and $v(s,a)$ is the sum across categories.

**Results on Pendragon Rising.** To validate that our findings generalize across different MACHIAVELLI games, we evaluate CCO on another game, *Pendragon Rising*.

*Table 8.* MACHIAVELLI results on *Pendragon Rising* (100 steps, 10 seeds).

| Method | Violations | Reward | $\max_t \lambda_t$ |
|---|---|---|---|
| Always Baseline | $0.0\%$ | $250.0 \pm 80.0$ | — |
| CCO $\alpha{=}0.05$ | $6.1 \pm 1.0\%$ | $316.2 \pm 60.9$ | $0.89 \pm 0.14$ |
| CCO $\alpha{=}0.10$ | $8.2 \pm 1.1\%$ | $332.2 \pm 41.6$ | $0.76 \pm 0.07$ |
| CCO $\alpha{=}0.15$ | $12.0 \pm 3.6\%$ | $343.3 \pm 42.5$ | $0.71 \pm 0.22$ |
| $\lambda = 0$ | $30.0 \pm 0.0\%$ | $435.0 \pm 0.0$ | — |

Results are consistent with *Hero of Kendrickstone*: CCO achieves strong calibration across all target levels, with $\alpha = 0.05$ yielding 6.1% violations, $\alpha = 0.10$ yielding 8.2%, and $\alpha = 0.15$ yielding 12.0%—all close to their respective targets. The pattern of decreasing $\max_t \lambda_t$ with looser targets again appears clearly ($0.89 \to 0.76 \to 0.71$), confirming that the controller adapts conservatism appropriately. The $\alpha = 0.10$ configuration reduces violations from 30.0% to 8.2% (a 73% reduction) while retaining 76% of unconstrained reward.

# H. Comparing CCO to AUP

The experiments in the main text demonstrate CCO's effectiveness on LLM-based agents operating in complex, realistic environments. However, these settings make it difficult to isolate the specific benefits of online calibration versus the fixed-penalty approach of Attainable Utility Preservation (AUP). To provide a controlled comparison, we design a gridworld environment that creates the conditions where the difference between adaptive and fixed conservatism becomes most apparent: non-stationary dynamics that shift the optimal safety-utility tradeoff over time.

The key insight motivating this experiment is that fixed-$\lambda$ methods implicitly assume a stationary relationship between the penalty magnitude and the resulting violation rate. When environmental conditions change—for instance, when the density of harmful states fluctuates seasonally—a fixed penalty weight becomes miscalibrated. It may be too permissive in high-risk

periods and overly conservative in low-risk periods. CCO's online calibration addresses this by continuously adjusting $\lambda_t$ based on observed outcomes, automatically adapting to the current risk level without requiring prior knowledge of the environmental dynamics.

## H.1. Environment Design

We construct a seasonal foraging gridworld as a controlled Markov process $(\mathcal{S}, \mathcal{A}, P)$ that captures the essential challenge of non-stationary safety constraints. The environment consists of a $10 \times 12$ grid where an agent navigates from a start position to a goal while avoiding harm to species that inhabit each cell. The critical feature is that species density varies with a temperature cycle that creates distinct seasons: summer periods with high species density (and thus high potential for harm) alternate with winter periods where species are sparse.

Formally, a state $s \in \mathcal{S}$ comprises the agent's grid position $(r, c)$, a species density tensor $\mathbf{G} \in \mathbb{R}_{\geq 0}^{10 \times 12 \times 8}$ tracking eight distinct species across all cells, a temperature scalar $T \in [0, 1]$, and a season index. The agent chooses from five actions $\mathcal{A} = \{\text{UP}, \text{DOWN}, \text{LEFT}, \text{RIGHT}, \text{STAY}\}$, with STAY serving as the baseline action $a_0$. Movement actions translate the agent by one cell (clipped at boundaries), while species populations evolve stochastically: spawning probability scales with temperature, and species die with fixed probability regardless of season.

The temperature follows a sinusoidal cycle $T_t = 0.5 + 0.5 \sin(2\pi t/L)$ with period $L = 50$ steps. This creates summer conditions ($T \approx 1$, high spawning) during $t \in [L/4, L/2]$ and winter conditions ($T \approx 0$, low spawning) during $t \in [3L/4, L]$. The non-stationarity is smooth and predictable in aggregate, but the agent does not observe the temperature directly—it must infer risk from the species densities it encounters.

## H.2. Objectives and Oversight Structure

The agent pursues a primary objective of reaching the goal cell $(9, 11)$ while minimizing harm to species. The primary scoring function rewards progress toward the goal via negative Manhattan distance, plus a bonus $R_{\text{goal}} = 10$ upon arrival:

$$U(s, a) = -d\big((r', c'), (9, 11)\big) + 10 \cdot \mathbf{1}[(r', c') = (9, 11)],$$

where $(r', c')$ denotes the position after taking action $a$. Meanwhile, harm is defined as the total species count at the destination cell: $\text{harm}(s, a) = \sum_{i=1}^{8} G_{r', c', i}$.

Following the AUP formulation, we implement auxiliary overseers as optimal Q-value functions for species-specific reward functions. Each auxiliary reward $R_i(s, a) = -G_{r', c', i}$ penalizes harm to species $i$, and the corresponding Q-values $q_i(s, a) = Q_{R_i}^*(s, a)$ are computed via value iteration with discount $\gamma = 0.99$. This setup directly instantiates the original AUP penalty structure within our CCO framework, enabling a fair comparison that isolates the effect of online calibration.

For the loss function, we use the binary indicator $\ell(s, a) = \mathbf{1}[\text{harm}(s, a) > 0]$, which records whether any species were harmed. This choice makes violation rates directly interpretable as the fraction of steps causing harm.

## H.3. Baseline: Fixed-$\lambda$ AUP

The fixed-$\lambda$ AUP baseline uses identical penalty structure but maintains a constant conservatism level throughout deployment:

$$a_t = \arg \max_{a \in C(s_t)} \big[ U(s_t, a) - \lambda \cdot \Delta(s_t, a) \big].$$

The challenge with this approach is selecting $\lambda$: too small permits excessive harm during high-density periods, while too large causes unnecessary waiting during low-density periods. In non-stationary environments, no single $\lambda$ value is appropriate for all conditions.

## H.4. Experimental Protocol

We evaluate both methods across a range of parameter settings, running 5 episodes of 1000 steps each per configuration with different random seeds. For each run, we record the violation rate (fraction of steps with $\ell(s_t, a_t) > 0$) and total reward $(\sum_{t=0}^{T-1} U(s_t, a_t))$. To visualize behavioral differences on identical environments, we also generate trajectory comparisons by initializing with a fixed seed, advancing to a target season (summer or winter), recording the species grid, and running both agents from position $(0, 0)$ for 70 steps on this shared snapshot.

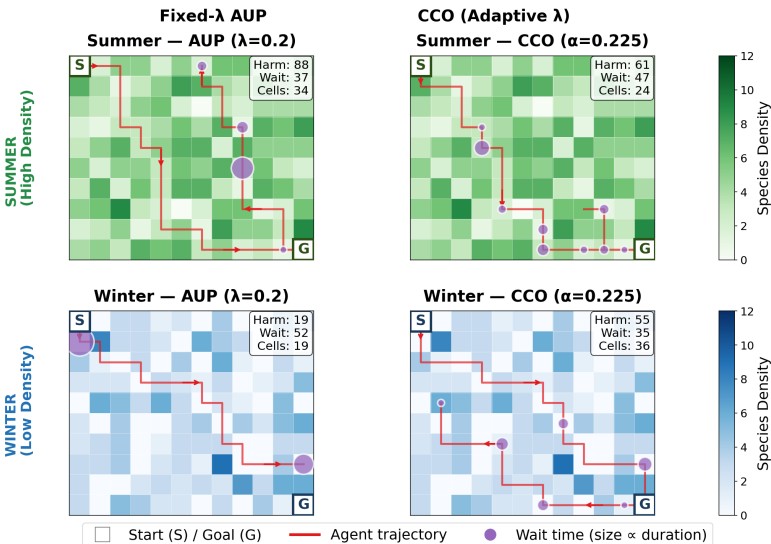

*Figure 2.* Agent trajectories (70 steps) across seasons. Purple circles indicate waiting, with size proportional to duration. Fixed-$\lambda$ AUP ($\lambda = 0.2$) exhibits excessive waiting regardless of season, while CCO ($\alpha = 0.225$) adapts its conservatism to current conditions while maintaining the target violation rate.

*Table 9.* Behavioral comparison across seasons (70-step trajectories on identical environments).

| Season | Agent | Harm | Max dwell | Cells visited | Notes |
|--------|-------|------|-----------|---------------|-------|
| Summer | AUP $\lambda$=0.2 | 88 | 24 | 34 | More harm; frequent waiting |
| Summer | CCO $\alpha$=0.225 | 61 | 14 | 24 | Less harm; less waiting |
| Winter | AUP $\lambda$=0.2 | 19 | 33 | 19 | Overly conservative (stalls) |
| Winter | CCO $\alpha$=0.225 | 55 | 12 | 36 | Adapts (explores more) |

## H.5. Results

Figure 2 and Table 9 reveal the fundamental difference between adaptive and fixed conservatism. In summer, when species density is high, both methods face genuine risk. However, AUP with $\lambda = 0.2$ accumulates more harm (88 vs. 61) while also waiting more (maximum dwell time 24 vs. 14 steps). This counterintuitive result occurs because AUP's fixed penalty weight is not calibrated to the current risk level—it waits when waiting is unnecessary and moves when moving is harmful.

The contrast is even starker in winter. With species sparse, the environment is objectively safer, yet AUP maintains the same conservatism level. The result is excessive caution: AUP waits for up to 33 consecutive steps and visits only 19 cells, while CCO recognizes the reduced risk, relaxes its conservatism, and explores 36 cells. CCO does incur more harm in winter (55 vs. 19), but this is the correct tradeoff: the harm is lower-stakes (fewer species per cell) and the additional exploration yields substantially more reward.

When we match methods at approximately equal violation rates (22.5% for AUP with $\lambda = 0.2$ versus 22.9% for CCO with $\alpha = 0.225$), the benefits of calibration become quantifiable. CCO achieves higher total reward (70 vs. 62) because it allocates its "violation budget" more efficiently—taking risks when they are worth taking and being cautious when caution is warranted. Perhaps more importantly, CCO exhibits dramatically lower variance in violation rates across seeds (standard deviation 0.1% vs. 1.3%), reflecting the stabilizing effect of the online controller.

