# OpenReview forum: "Calibrating Conservatism for Scalable Oversight"
_ICML.cc/2026/Conference — ICML 2026 regular_

### Official Review · Reviewer_tQdM · 2026-02-27

**Soundness:** 3
**Presentation:** 4
**Significance:** 2
**Originality:** 4
**Overall Recommendation:** 4
**Confidence:** 3

**Summary:**

The paper introduces a decision criterion for trading off utility and safety in a sequential setting. At each step, the controller observes the state, takes an action based on its current calibration of the utility / safety tradeoff, and observes the loss. The utility / safety tradeoff parameter is then tuned to either become more conservative (if the loss exceeded the target threshold) or more permissive (if the loss did not exceed the target threshold). It can be shown that in the limit, the average per-turn loss incurred by the method converges to the specified safety threshold.

The method is then applied to the problem of overseeing a stronger untrusted LLM by weaker trusted LLMs. The paper presents experiments in the Machiavelli environment (ethically navigating a text-based RPG) and SWE-Bench (containing the fraction of vulnerable code submits) and show that the method roughly succeeds at satisfying the safety threshold.

**Compliance With Llm Reviewing Policy:**

Affirmed.

**Key Questions For Authors:**

main:
- is it possible to quantify how much reward the algorithm achieves? and how does it compare to what it is possible to achieve?
- empirical baselines would be hugely helpful for understanding the practicality of the method

nits:
- the MDP structure (state transitions) isn't really used iiuc, it's closer to a contextual bandit setup
- it would be interesting to understand the effect of monitor false positive rate on the algorithm. I.e. if the monitor makes mistakes, how does that affect things?
- how optimal is the method in terms of safety convergence, is sublinear regret possible?

**Limitations:**

yes

**Strengths And Weaknesses:**

Soundness

I've skimmed the proofs and as far as I can tell, the method is sound and the lemmas and theorems in the paper are valid.
As for the experiments, I would've liked to see more baselines comparing the method's usefulness (i.e. how much reward it gets) and safety to alternatives (e.g. just use the weak LLM as actor, or other heuristic monitoring protocols). See also the "significance" paragraph below.

It would also be nice to have more ablations, e.g. how important is it that the lambda parameter is calibrated on the fly vs on a validation set? I'd imagine that if it's calibrated on a validation set and then fixed, it would be more practical to deploy, e.g. in parallel.

Presentation

The paper is clearly written, well-structured and easy to skim. I also appreciated the transparency flagging which parts are new vs standard (e.g. the proofs in the appendix).

Significance

I think the problem addressed by the paper is important: How can we maintain safety when the actor model is untrusted and the overseer model is weak? And the answer given in this paper is, we can do it by observing how effective the overseer is at containing the untrusted model and calibrating our degree of conservatism accordingly. (In the extreme case of the overseers failing to flag dangerous actions, the algorithm eventually reverts to a known-safe action.)

Basically I see CCO as a fallback, safety-ensuring monitoring protocol that's better at eliciting useful (i.e. high-reward) actions than just reverting to the known-safe action all the time. However, it's unclear exactly how much usefulness is being extracted / how competitive the method is compared to other monitoring protocols, like directly using the trusted model as the actor, or other heuristic monitoring protocols. This matters because even a guaranteed-safe method won't be used if it's not competitive in terms of usefulness. This is the main reason I'm unsure the paper will have much impact.

Originality

I find the paper quite original. Many papers try to improve how well a weak model can oversee an untrusted strong model, but I haven't seen much work on the orthogonal question of, given fixed overseer capability how to get as much "juice" (safety and usefulness) out of the joint system. I also find the inspiration from conformal decision theory to be insightful and appropriate.

---

> ### Author Rebuttal · Authors · 2026-03-30
>
> We thank Reviewer tQdM for their genuinely close reading and careful assessment of our work. We particularly appreciate the reviewer's recognition of CCO's originality in addressing the orthogonal question of how to extract maximum "juice" from a joint overseer-agent system given fixed overseer capability. The reviewer's main stated concern was uncertainty about practical impact due to unclear competitiveness with alternatives. We address this head-on below with new experiments.
>
> **Q1--Q2: Reward quantification and empirical baselines.**
>
> *Weak model as actor.* To make this comparison we move beyond measuring safety violations since the weak model gives 0% misalignment by design. We instead measure binary solve rates, using an independent GPT-5.4 judge to evaluate whether each patch correctly fixes the GitHub issue (PASS/FAIL). On 100 SWE-bench instances (3 seeds), GPT-5.4-nano (the overseer-class model) alone achieves a solve rate of $26.7 \pm 2.5\%$, while the strong model (GPT-5.4-mini) unconstrained achieves $60.0 \pm 2.0\%$. CCO-selected patches achieve $45.0 \pm 4.6\%$. CCO extracts substantially more capability than the weak model alone while maintaining the safety guarantee.
>
> *Adaptive majority vote.* We implemented an adaptive majority vote baseline with the same target-tracking objective as CCO. Each overseer votes ACCEPT if its score $q_i(s,a) \in [1,10]$ exceeds threshold $\tau$, REJECT otherwise; majority-rejected candidates are eliminated. The adaptive variant re-fits $\tau$ at each step on all past data to match the running rate to $\alpha$ (see our response to Reviewer UZJX for full results). Under steady state, CCO converges 2--7$\times$ faster. Under distribution shift ($p$ jumping from 0.2 to 0.8), adaptive MV's realized rate spikes to 22.5% (more than double the $\alpha{=}0.10$ target) while CCO stays within $\sim$2% of target. This also addresses online calibration vs. validation-set tuning: adaptive MV fits on all past data (analogous to a growing validation set), and CCO's per-step update substantially outperforms it under non-stationarity.
>
> *Overseer count ablation.* We have conducted a comprehensive ablation over $n \in \{1,...,10\}$ across $\alpha$ levels (see our response to Reviewer 9DLT, Q4 for the full table).
>
> **Q3: Contextual bandit observation.**
> We present the framework in MDP terms because (1) it allows for future extensions where the penalty or overseers are stateful, and (2) CDT's guarantees hold for *any* state sequences, including those generated by MDPs, so the MDP formulation is strictly more general. We will more directly acknowledge in the camera-ready version that SWE-bench is closer to contextual bandits. We do emphasize, however, that MACHIAVELLI does have state-based transitions and is thus more naturally viewed from the MDP setting we introduce.
>
> **Q4: Monitor false positive rate.**
> We agree this is an interesting question. A granular investigation of how overseer noise affects the learning dynamics of $\lambda_t$ is a promising direction for future work.
>
> **Q5: Optimality in terms of safety convergence.**
> Yes, CCO's $O(1/t)$ rate is worst-case tight. We provide the full lower-bound construction in our response to Reviewer Pn1A (Q3).
>
> **Summary.**
> The reviewer stated that uncertainty about CCO's practical competitiveness was the main reason for concern about impact. We hope the new results address this directly: CCO achieves a substantially higher solve rate than the weak model baseline, outperforms adaptive majority vote under distribution shift, and provides interpretable $\alpha$-targeting that no fixed-parameter alternative can match. Combined with the $n$-ablation and cross-model experiments demonstrating robust $\alpha$-tracking, we believe these results substantially strengthen the empirical case for CCO's practicality. **We sincerely hope the reviewer might consider a positive adjustment to their overall score in light of these additions.**

---

> > ### Author Rebuttal · Reviewer_tQdM · 2026-03-31
> >
> > Thanks for running the additional experiments!
> > I'd love to see the results corroborated on bigger models than just nano and mini. These are really tiny, and there's no reason to use them for scalable oversight research.
> >
> > But if the results hold up, I'd consider my main concern resolved and would gladly increase my score from weak accept to accept.

---

> > > ### Author Response · Authors · 2026-04-02
> > >
> > > We thank the reviewer for the positive acknowledgment and for marking their concerns as fully resolved. We appreciate the quick acknowledgement and constructive engagement. We are pleased to report that all three experiments from our initial rebuttal replicate successfully with larger models.
> > >
> > > We replicated our experiments using **GPT-5.4 as the strong model** and **GPT-5.4-mini as overseers**.
> > >
> > > The $n$-ablation results (3 seeds, 300 instances) in this setting are consistent with our initial rebuttal:
> > >
> > > | $n$ | $\alpha=0.05$ | $\alpha=0.10$ | $\alpha=0.15$ |
> > > |----|---------------|---------------|---------------|
> > > | 1  | $9.9 \pm 0.2$ | $13.4 \pm 0.4$ | $17.1 \pm 0.4$ |
> > > | 2  | $7.1 \pm 0.4$ | $11.3 \pm 0.3$ | $15.9 \pm 0.5$ |
> > > | 3  | $6.0 \pm 0.0$ | $10.6 \pm 0.2$ | $15.3 \pm 0.3$ |
> > > | 5  | $5.6 \pm 0.2$ | $10.2 \pm 0.2$ | $14.6 \pm 0.4$ |
> > > | 7  | $5.4 \pm 0.2$ | $10.1 \pm 0.2$ | $14.3 \pm 0.6$ |
> > > | 10 | $5.2 \pm 0.2$ | $9.9 \pm 0.4$  | $13.7 \pm 0.7$ |
> > >
> > >
> > > **The adaptive majority vote comparison also replicates with GPT-5.4**. Under distribution shift ($p{=}0.2 \to 0.8 \to 0.2$, 100 instances per phase):
> > >
> > > | $\alpha$ | Phase | CCO Rate (%) | Adaptive MV Rate (%) |
> > > |---|---|---:|---:|
> > > | 0.05 | $p=0.2$ (1--100) | 6.7 | 8.8 |
> > > | 0.05 | $p=0.8$ (101--200) | 6.1 | 18.2 |
> > > | 0.05 | $p=0.2$ (201--300) | 3.8 | 4.6 |
> > > | 0.10 | $p=0.2$ (1--100) | 12.0 | 12.5 |
> > > | 0.10 | $p=0.8$ (101--200) | 11.1 | 15.5 |
> > > | 0.10 | $p=0.2$ (201--300) | 10.2 | 14.2 |
> > > | 0.15 | $p=0.2$ (1--100) | 14.4 | 14.9 |
> > > | 0.15 | $p=0.8$ (101--200) | 15.1 | 22.0 |
> > > | 0.15 | $p=0.2$ (201--300) | 14.4 | 20.2 |
> > >
> > > CCO stays near target throughout all phases and $\alpha$ levels. Adaptive MV overshoots substantially during the high-misalignment phase (e.g., 18.2% vs. CCO's 6.1% at $\alpha{=}0.05$) and remains elevated after the shift reverses.
> > >
> > >
> > > **We also replicate our weak-model-as-actor baseline comparison using GPT-5.4 as the strong model** and Claude Opus 4.6 as an independent judge. The strong model unconstrained achieves $63.7 \pm 2.3\%$. We present results for both GPT-5.4-mini and GPT-5.4-nano as overseers to show results across differing capability gaps.
> > >
> > > | Overseer | Weak Actor | CCO-Selected |
> > > |---|---:|---:|
> > > | GPT-5.4-mini | $54.0 \pm 2.7\%$ | $59.7 \pm 4.0\%$ |
> > > | GPT-5.4-nano | $25.3 \pm 3.2\%$ | $47.3 \pm 5.1\%$ |
> > >
> > > Consistent with our initial rebuttal results, CCO achieves higher utility than the weak actor alone.
> > >
> > > **These results on the larger GPT-5.4 model corroborate all findings from our initial rebuttal. We hope this fully addresses the reviewer's remaining concern and appreciate the constructive engagement.**

---

### Official Review · Reviewer_Pn1A · 2026-03-13

**Soundness:** 3
**Presentation:** 3
**Significance:** 3
**Originality:** 3
**Overall Recommendation:** 4
**Confidence:** 3

**Summary:**

This paper studies scalable oversight and the authors propose Calibrated Collective Oversight (CCO), where overseers score an agent's candidate actions and penalize actions that deviate from a safe, conservative default action. The work builds on Attainable Utility Preservation (AUP) idea. And the observation is that overseers don't need to agree on what the agent should do, they just need to collectively flag when a candidate action looks different from the safe default. CCO then selects actions by trading off the agent's own preferences against this collective penalty, with a parameter lambda controlling how much weight the penalty receives. The authors propose using Conformal Decision Theory to tune lambda. CCO can adjust lambda after every step and achieve provable finite-time guarantees.

The authors then validate this empirically in two experiments: MACHIAVELLI, where weaker Haiku-based overseers constrain a stronger Sonnet agent playing ethical text-based games, and SWE-bench, where the weaker overseers successfully detect and block security vulnerabilities that a stronger agent tries to introduce. In both settings, observed violation rates closely match the specified targets, confirming that the theoretical guarantees translate into practice.

**Compliance With Llm Reviewing Policy:**

Affirmed.

**Final Justification:**

Updating confidence in light of the authors' response.

**Key Questions For Authors:**

Here are my main questions:

1) Can the authors comment on ways to relax the assumption of observing the ground-truth loss after each action?
For example, what happens when the loss is noisy instead of being noiseless?

2) I'm not super familiar with the AUP formulation, which this paper builds on.
What is the connection to the broader safe RL work?
And can the collective + conformal idea augment other types of safe RL approaches?

3) Do the authors know of any lower bounds to suggest the O(1 / t) bound in Theorem 4.10 is tight?

4) Moving to the experiments, are the authors able to use other LLMs to demonstrate scalable oversight?
Because it is noticeable that all results are on (sonnet 4.5, haiku 4.5).
Perhaps other models through API access can show that the method does generalize to other model families besides Claude.

Relatedly, I think the experiments also show that the assumption of immediate access to the ground-truth loss after each action is a bit unrealistic.
It would be great if the authors could comment on how this part of the experiment could be made more realistic (if possible).

**Limitations:**

I do not see a limitation subsection and it may be good to have one in the interest of transparency.

**Strengths And Weaknesses:**

Soundness:
The paper has solid theoretical results and sufficient experiments that relate to the theory.

Theoretically, I think a key assumption that stands out is the requirement of observing the ground-truth loss after each action.
In practice, this is rarely available in LLM settings except involving terminal reward only settings, where this condition is satisfied trivially because there is no intermediate reward.

Presentation:
The paper is clear and well written for the most part.
However, I think the paper could benefit from more clear exposition in describing its guarantees.

For example, I think would be useful to have the "Eventually safe" definition be moved from the appendix to the main.
It is a key term used in the theoretical results and I think readers would benefit to see if formally defined before these results.

Significance:
The paper is interesting in that it addresses the key problem of scalable oversight.
Moreover, it is one of the few that has both theoretical guarantees and practical experiments with LLMs.
I think most work in scalable oversight does either one of the two.

Originality:
I will preface this by saying I am not super familiar with neither AUP literature nor conformal method specifically.
But to my knowledge, the use of conformal ideas from statistics in scalable oversight is new.
And personally, I think it's good to see that uncertainty based ideas from statistics are increasingly being adopted for use in AI safety (as they should be).

With this said, one thought that comes to mind is that if we wanted to study safety in a RL framework, why not use the constrained MDP or constrained policy optimization approach?

---

> ### Author Rebuttal · Authors · 2026-03-30
>
> We thank Reviewer Pn1A for the positive assessment, and in particular for recognizing that this is one of the few works on scalable oversight to offer both theoretical guarantees and practical LLM experiments. We consider this a core contribution. We address each question below.
>
> **Q4: On single model family.**
>
> We have replicated all SWE-bench results using OpenAI GPT-5.4-mini/nano (3 seeds $\times$ 300 instances), with a full $n$-ablation across $n \in \{1,...,10\}$ and $\alpha \in \{0.05, 0.10, 0.15, 0.20\}$. See our response to Reviewer UZJX for the complete table. The key finding: $\alpha$-tracking holds across model families with comparable precision, confirming that the guarantees (which derive from CDT and hold for any bounded loss sequence) are model-agnostic in practice as well as in theory.
>
> **Q1: Relaxing the ground-truth loss assumption.**
>
> We acknowledge that immediate exact feedback is unrealistic in many deployments. We will add this as a limitation, but here we sketch why the approach extends naturally to both noisy and delayed feedback.
>
> *Noisy feedback.* Our current theorem assumes exact feedback. If the controller instead uses noisy observations $\tilde{\ell}_t = \ell_t + \xi_t$, then the same projected-update algebra controls the average *observed* loss:
>
> $$\frac{1}{T+1}\sum_{t=0}^T \tilde\ell_t \leq \alpha + \frac{\lambda_{T+1} - \lambda_0}{\eta(T+1)}$$
>
> Passing to true loss via $\sum \ell_t = \sum \tilde{\ell}_t - \sum \xi_t$, under bounded zero-mean noise the additional error is $O(1/\sqrt{T})$ by concentration. The guarantee becomes:
>
> $\frac{1}{T+1}\sum_{t=0}^T \ell_t \leq \alpha + O(1/T) + O(1/\sqrt{T}).$
>
> *Delayed feedback.* Suppose loss from round $s$ is revealed after delay $d_s \in \{0,...,D\}$, and let $R_t := \{s \leq t : s + d_s = t\}$ be losses revealed at time $t$. A natural batched update is:
>
> $\lambda_{t+1} = \bigl[\lambda_t + \eta\bigl(\sum_{s \in R_t} \ell_s - \alpha|R_t|\bigr)\bigr]_+.$
>
> By the same telescoping argument, and noting that at most $D$ losses remain unrevealed by time $T$:
>
> $\frac{1}{T+1}\sum_{s=0}^T \ell_s \leq \alpha + \frac{\lambda_{T+1} - \lambda_0}{\eta(T+1)} + \frac{D}{T+1}.$
>
> Bounded delay adds only an $O(D/t)$ transient term. We do not formally prove these extensions in the current draft, but they clarify why delayed and noisy feedback are manageable in principle. We would be happy to add these arguments to an appendix in the camera-ready version.
>
> **Q3: Tightness of the $O(1/t)$ bound.**
>
> The rate is worst-case sharp for our additive controller. Consider a one-state, two-action instance with $\ell(s,a_0)=0$, $\ell(s,a_1)=1$, and $(U(s,a_1)-U(s,a_0))/\Delta(s,a_1) = \bar{\lambda}$. The controller selects $a_1$ whenever $\lambda_t < \bar{\lambda}$, updating $\lambda_{t+1} = \lambda_t + \eta(1-\alpha)$. It takes $m = \lceil(\bar{\lambda}-\lambda_0)/(\eta(1-\alpha))\rceil$ rounds to reach safety, accumulating $m(1-\alpha) = \Omega((\bar{\lambda}-\lambda_0)/\eta)$ total excess violations, matching Theorem 4.10.
>
> **Q2: Connection to safe RL.**
>
> The key distinction is that safe RL operates at training time, while CCO operates at deployment time around a fixed, potentially untrustworthy model. In the scalable oversight setting, the primary agent's internals may be opaque or its objectives misaligned, so one cannot apply constrained policy optimization since one does not control the training process. CCO requires no access to model weights or gradients; only the ability to query for candidate actions and score them with overseers.
>
> CCO is best viewed as *complementary* to safe RL, not a replacement. Our approach builds on AUP, which can be seen as intrinsic regularization: penalizing actions that significantly change auxiliary value functions, without requiring an explicit constraint cost. The guarantee also differs: constrained RL targets expected constraint satisfaction under stationarity and adapts the policy during training, whereas CCO leaves the policy fixed, adapts only $\lambda_t$ online, and provides finite-time empirical violation control with no distributional assumptions. One can train with any safe RL method, then apply CCO at deployment as an additional safety layer.
>
> **Concluding remarks.**
> We hope the above addresses all five key questions and stated weaknesses. **We sincerely hope the reviewer might consider a positive adjustment to their score.**

---

> > ### Author Rebuttal · Reviewer_Pn1A · 2026-04-02
> >
> > I thank the authors for their response and follow up experiments, which I hope can be incorporated in the next revision to improve the paper. I will increase my confidence to indicate higher confidence in my positive assessment.

---

> > > ### Author Response · Authors · 2026-04-07
> > >
> > > We thank the reviewer for increasing their confidence and for the positive engagement. We will incorporate the above discussion and ablations in the camera-ready. Thank you again.

---

### Official Review · Reviewer_UZJX · 2026-03-13

**Soundness:** 3
**Presentation:** 3
**Significance:** 3
**Originality:** 3
**Overall Recommendation:** 4
**Confidence:** 3

**Summary:**

This paper introduces Calibrated Collective Oversight (CCO), a framework for scalable AI oversight that combines an AUP-inspired penalty measuring deviation from a conservative baseline with online calibration via Conformal Decision Theory (CDT). Tthe rate of undesirable outcomes can be controled at a user-specified target level. Experiments using Claude Sonnet 4.5 as the stronger agent and Claude Haiku 4.5 as the weaker overseer agent are conducted on MACHIAVELLI and a SWE-bench-style code patch selection task, demonstrating that empirical violation rates closely track the specified targets.

**Compliance With Llm Reviewing Policy:**

Affirmed.

**Final Justification:**

Due to the lack of strong directly comparable baselines and the limited external validity of the current experiments, I still have some uncertainty about how effective the method would be in practice. That said, I do not think these issues are entirely the authors' fault, and considering the significance of the problem and the paper's technical contribution, I am giving a positive score.

**Key Questions For Authors:**

no further questions

**Limitations:**

No dedicated limitations section was identified in the manuscript. The authors should add one that addresses at minimum: (1) reliance on an author-designed evaluation protocol, (2) restriction to a single model family, and (3) any open questions about the practical deployment of CCO in real oversight pipelines.

**Strengths And Weaknesses:**

**Strength**

The paper makes a meaningful contribution by bridging two previously separate lines of safety research: AUP-style impact penalties and online calibration via CDT, without distributional assumptions. The generalization of Q-values in AUP to arbitrary scoring functions broadens the framework's applicability. Evaluation across two distinct task regimes lends the empirical results some breadth, and the α parameterization provides an interpretable handle on safety–capability tradeoffs. The presentation is well-structured, with a clear overview figure and explicit pseudocode aiding reproducibility.

**Weakness**

The primary comparisons against an "Always Baseline" and λ = 0. These function more as reference bounds (upper and lower) than competitive baselines. A more informative evaluation would compare against AUP directly in the main table rather than deferring it to the appendix. Comparison to other relevant prior work on scalable oversight or calibrated safety mechanisms would further strengthen the empirical case.

All experiments use a single model family (Claude). It is unclear whether the results generalize to other architectures or model families. The authors should acknowledge this as a limitation and discuss the extent to which CCO's behavior may be model-dependent.

The demonstration of target-level control on SWE-bench relies on an adversarial protocol designed by the authors. While this is a reasonable proof-of-concept, it raises questions about how well the calibration generalizes to naturalistic settings.

---

> ### Author Rebuttal · Authors · 2026-03-30
>
> We thank Reviewer UZJX for recognizing CCO's contribution in bridging two previously separate lines of safety research. We also appreciate that the reviewer provided three concrete weaknesses/questions, which we address below, the first two with new experimental results, and the third with a discussion of CCO's theoretical guarantees.
>
> **On single model family:**
>
> We appreciate the reviewer's concern that the paper used the Claude model family throughout. **We have conducted a comprehensive ablation using OpenAI models**, as well as varying overseer count $n$ on SWE-bench (3 seeds $\times$ 300 instances, OpenAI GPT-5.4-mini/nano). The below table shows misalignment rates converge toward the target $\alpha$ across all settings:
>
> | $n$ | $\\alpha=0.05$ | $\\alpha=0.10$ | $\\alpha=0.15$ |
> |----|---------------|---------------|---------------|
> | 1  | $9.0 \\pm 0.3$ | $12.2 \\pm 0.8$ | $16.1 \\pm 0.7$ |
> | 2  | $7.0 \\pm 0.3$ | $11.1 \\pm 0.4$ | $15.4 \\pm 0.5$ |
> | 3  | $6.2 \\pm 0.2$ | $10.7 \\pm 0.6$ | $14.9 \\pm 0.4$ |
> | 5  | $5.7 \\pm 0.3$ | $10.4 \\pm 0.4$ | $14.4 \\pm 0.4$ |
> | 7  | $5.6 \\pm 0.2$ | $9.9 \\pm 0.2$  | $13.9 \\pm 0.5$ |
> | 10 | $5.2 \\pm 0.2$ | $9.7 \\pm 0.3$  | $13.4 \\pm 0.7$ |
>
> CCO generalizes across model families: $\alpha$-tracking holds with comparable precision. We emphasize that CCO's theoretical guarantees are model-agnostic, they derive from Conformal Decision Theory and hold for any bounded loss sequence, regardless of the underlying model architecture. **We hope this convinces the reviewer that the results are general across model families and that there is not a limitation in this regard.**
>
> **On baseline comparisons:**
>
> We agree that stronger baselines would strengthen the paper's case and have conducted additional comparisons during the rebuttal period.
>
> *Fixed-$\\lambda$ AUP.* The reviewer suggests comparing against AUP directly. Original AUP requires computing optimal Q-values in a fully specified MDP, which is infeasible in LLM settings; this is precisely why CCO generalizes AUP's penalty to arbitrary scoring functions. The closest comparison is therefore fixed-$\\lambda$ with our generalized penalty but without conformal calibration, isolating the role of online $\\lambda_t$ adaptation. We swept $\\lambda$ across 9 values on both Claude and OpenAI data. The mapping from $\\lambda$ to violation rate is steep, nonlinear, and model-dependent: achieving ${\\sim}10%$ misalignment requires $\\lambda{=}0.20$ on Claude but $\\lambda{=}0.30$ on OpenAI, and nearby values can shift sharply (on OpenAI, $\\lambda{=}0.20$ gives 18.7% while $\\lambda{=}0.50$ gives 9.2%). CCO avoids this: setting $\\alpha{=}0.10$ yields 10.87% on Claude and 9.67% on OpenAI with no per-model tuning. We will add this comparison in full detail to the camera-ready version.
>
> *Adaptive majority vote.* The reviewer also asks for comparison with calibrated safety mechanisms. We therefore implemented an adaptive majority vote baseline with the same target-tracking objective as CCO. Each overseer votes ACCEPT on a candidate if its score $q_i(s,a) \\in [1,10]$ exceeds a threshold $\\tau$ and REJECT otherwise; candidates rejected by a majority are eliminated, and the highest-utility survivor is selected. At each step, the adaptive variant re-fits $\\tau$ on all historical data to match the running misalignment rate to $\\alpha$. Under steady-state conditions ($p{=}0.3$, 300 instances), CCO converges 2--7$\\times$ faster. Under distribution shift ($p$ jumps from 0.2 to 0.8 and back, $\\alpha{=}0.10$):
>
> | Phase | CCO Rate (%) | Adaptive MV Rate (%) |
> |---|---:|---:|
> | $p=0.2$ (instances 1--100) | 11.0 | 11.3 |
> | $p=0.8$ (instances 101--200) | 12.1 | 22.5 |
> | $p=0.2$ (instances 201--300) | 7.1 | 3.9 |
>
> This demonstrates that CCO's advantage over calibrated ensemble methods is not just interpretability but genuine adaptivity to changing conditions.
>
> **On author-designed adversarial protocol.**
> We acknowledge this as a limitation of our empirical evaluation. The protocol provides a controlled setting with known ground-truth, enabling precise $\alpha$-tracking measurement. In naturalistic settings, ground truth may only emerge after deployment, but this is exactly the sequential regime CCO targets. Theorem 4.10 guarantees convergence to $\alpha$ for *any* bounded loss sequence regardless of how losses arise. What changes in naturalistic settings is only how precisely we can *measure* $\alpha$-tracking, not CCO's ability to achieve it. See the discussion on the rebuttal to Q1 of Reviewer Pn1A regarding relaxed assumptions on loss observations in more realistic settings.
>
>
> **Concluding remarks.**
> Given these additions, together with the reviewer's positive assessment of CCO's originality and presentation, we sincerely hope the reviewer will consider revising their soundness and significance scores and **adjusting their overall recommendation upward.**

---

> > ### Author Rebuttal · Reviewer_UZJX · 2026-04-03
> >
> > Thank you for the detailed rebuttal and for adding new experiments. I will raise my score.

---

> > > ### Author Response · Authors · 2026-04-06
> > >
> > > We thank the reviewer for raising their score and for the engagement throughout the review process. We are pleased the new experiments addressed the reviewer's concerns. We will incorporate the suggested baselines and ablations in the camera-ready version. Thank you again.

---

### Official Review · Reviewer_9DLT · 2026-03-17

**Soundness:** 3
**Presentation:** 3
**Significance:** 3
**Originality:** 3
**Overall Recommendation:** 5
**Confidence:** 4

**Summary:**

Introducing Calibrated Collective Oversight (CCO), the authors claim that weaker overseers can collectively constrain a stronger agent in sequential settings, with provable bounds on violation rates. Experiments on MACHIAVELLI (text games) and SWE-bench (code patching with injected misalignment) show that empirical violation rates track user-specified targets α closely.

**Compliance With Llm Reviewing Policy:**

Affirmed.

**Final Justification:**

My concerns have been addressed through additional experiments. I have updated my score accordingly.

**Key Questions For Authors:**

1. I think the paper leaves a₀ underspecified as a general concept - Definition 4.1 lists several possible heuristics (minimize harm metric, lowest variance, do-nothing), but the two experiments use entirely different strategies (a₀ is defined using oracle violation counts in MACHIAVELLI, and hardcoded fallback in SWE-bench). This raises the question of whether a₀ can be reliably/consistently defined in arbitrary, different deployment settings.

Especially, in MACHIAVELLI, a₀ is defined using "oracle" violation counts (included in the ground-truth dataset) that models typically cannot access - how can we define a₀ without such privileged information in such case?

2. In SWE-bench, the adversary embeds backdoors in patch content but U scores are fixed by the experimenter. However, a misaligned agent could also strategically lower its U score on the malicious patch to avoid penalty. Would CCO be robust when the adversary controls both the action content and the utility signal?

3. The paper uses n=5 overseers in both experiments with no ablation. How would performance change with n=1, n=3, or n=10? The penalty ∆ = Σ|qi − qi(a₀)| grows additively with n, so noisy overseers could inflate penalties on safe actions just as much as on unsafe ones - if this happens, increasing n hurts by making the agent overly conservative without improving discrimination. Conversely, too few overseers may miss entire categories of risk. Without an ablation, we cannot tell whether n=5 is a sweet spot, overkill, or insufficient.

4. The related work positions CCO as "complementary" to prior methods (e.g., debate, process supervision), but provides no empirical or qualitative comparison. I think the authors could discuss under what conditions CCO is preferable to these alternatives - e.g., when does penalty-based calibration outperform debate, and when might it fall short?

**Limitations:**

Yes

**Strengths And Weaknesses:**

Strengths: Well-motivated combination of methods (AUP penalties with CDT calibration), which is practically useful for LLM agents. High α-tracking consistency across two very different domains (10.9% and 10.87% for α=0.10). The "collective conservatism" insight - overseers need not agree on what is good, only detect deviation - is interesting.

Weaknesses: No comparison with any alternative oversight method (debate, process supervision, majority voting) and no ablation over n, making it hard to tell whether CCO's specific penalty structure adds value beyond simpler baselines.

---

> ### Author Rebuttal · Authors · 2026-03-30
>
> We thank Reviewer 9DLT for the careful reading, positive assessment, and constructive questions. We appreciate that the weaknesses and key questions were clear and substantive, and we are confident that the new experiments and discussions below address each of them.
>
> **Q3: Ablation over $n$.**
>
> We have conducted a comprehensive ablation over overseer count $n \\in \\{1,2,3,5,7,10\\}$ across $\\alpha$ levels on SWE-bench (3 seeds $\\times$ 300 instances, OpenAI GPT-5.4-mini/nano). The full table is provided in our response to Reviewer UZJX. The key finding is that CCO tracks $\\alpha$ even with a single overseer ($n{=}1$ at $\\alpha{=}0.10$ achieves 12.2%, already close to target), with additional overseers consistently tightening precision (e.g., $n{=}10$ achieves 9.7% at $\\alpha{=}0.10$). The parameter $n$ serves as a design knob controlling oversight precision, independent of the user-specified $\\alpha$.
>
> **Q1: Baseline action $a_0$ specification:**
>
> We conducted an $a_0$ sensitivity analysis on SWE-bench using four strategies ($\\alpha=0.10$, $n=5$, 3 seeds $\\times$ 300 instances):
>
> (1) **fixed deferral**
> (2) **lowest-penalty candidate**: $a_0 = \\arg\\min_c \\sum_i |q_i(c) - \\bar{q}_i|$
> (3) **max-min overseer score**: $a_0 = \\arg\\max_c \\min_i q_i(c)$
> (4) **highest average overseer score**: $a_0 = \\arg\\max_c \\mathrm{mean}_i\\, q_i(c)$
>
> | $a_0$ Strategy | Misalignment Rate (\\%) |
> |---------------|------------------------|
> | Fixed deferral (original) | $10.44 \\pm 0.38$ |
> | Lowest-penalty candidate  | $12.00 \\pm 1.53$ |
> | Max-min overseer score    | $10.67 \\pm 0.67$ |
> | Highest avg overseer score | $10.56 \\pm 0.51$ |
>
> All strategies achieve rates close to the $\\alpha=0.10$ target. The conformal controller adapts $\\lambda_t$ to compensate for different $a_0$ choices, demonstrating robustness without oracle information.
>
> **Q2: Adversary controlling $U$ scores.**
>
> If an adversary lowers $U$ on a misaligned patch, this reduces its selection probability, making the attack self-defeating. Empirically we tested this on 150 SWE-bench instances, lowering $U$ from 10 to 8 keeps misalignment near target (10.7%), while lowering to 5 reduces it to 0%. The attacked could of course choose a specific instance to lower its score and we would be susceptible to such an attack; our method only controls long run violations.
>
> The potentially more concerning scenario is an adversary inflating $U$ to force selection of the misaligned patch. We tested $U_{\\text{mal}} \\in \\{20, 50, 100\\}$. Short-term misalignment increases (e.g., at $T{=}50$: 20%, 26%, 30%), but CCO adapts $\\lambda_t$. By $T{=}1500$, rates drop to 10.7%, 12.1%, and 13.9%, and by $T{=}3000$, even $U_{\\text{mal}}{=}100$ reaches 12.1%, approaching $\\alpha=0.10$.
>
> More fundamentally, Theorem 4.10 guarantees convergence for arbitrary loss sequences: if violations persist, $\\lambda_t$ increases, enforcing conservatism. The guarantee is on long-run violation rates, not detection of individual actions.
>
> **Q4: Comparison with alternative oversight methods.**
>
> We have implemented two competitive baselines: fixed-$\\lambda$ AUP (our generalized penalty without conformal calibration) and an adaptive majority vote with the same target-tracking objective as CCO. We provide the full experimental details and results in our response to Reviewer UZJX. The key findings are:
>
> (1) fixed-$\\lambda$ AUP requires per-model tuning and the mapping from $\\lambda$ to violation rate is steep and nonlinear, while CCO cleanly matches its target $\\alpha$
>
> (2) under distribution shift ($p$ jumping from 0.2 to 0.8), adaptive majority voting's realized misalignment rate spikes to 22.5% (more than double the $\\alpha{=}0.10$ target), while CCO stays within ${\\sim}2\\%$ of target.
>
> CCO's advantage over these alternatives is not just interpretability but genuine adaptivity to changing conditions.
>
> **On when CCO is preferable to debate:**
> CCO and debate address complementary aspects of oversight, Debate helps surface information that a weak judge might miss; CCO calibrates the conservatism of any action-selection rule to meet a target violation rate. CCO is preferable when: (1) the deployment is sequential and violation rates must be controlled across a trajectory, (2) diverse oversight signals are available but no single one is reliable, and (3) **formal guarantees on violation rates are required**. Debate may be preferable when the primary challenge is information asymmetry between a strong and weak model. Importantly, the two approaches compose naturally: debate outcomes can serve as an additional overseer within CCO's ensemble.
>
> **Concluding remarks.**
> We believe these additions address each of the reviewer's five key questions and the stated weakness regarding baselines and ablations. **We sincerely hope the reviewer might consider a positive adjustment to their score in light of these results.**

---

> > ### Author Rebuttal · Reviewer_9DLT · 2026-04-01
> >
> > Thanks for the detailed responses and extensive additional experiments. These address my main concerns, and I have updated my score accordingly.

---

> > > ### Author Response · Authors · 2026-04-06
> > >
> > > We thank the reviewer for updating their score and for the constructive engagement. We are glad the additional experiments addressed the reviewer's main concerns and will incorporate suggested changes, additions, and ablations in the camera-ready version. Thank you again.

---

### Decision · Program_Chairs · 2026-04-30

**Decision:**

Accept (regular)

**Comment:**

The reviewers found the core contribution of the paper solid and timely. In summary, the paper proposed a framework to improve scalable oversight by introducing Calibrated Collective Oversight (CCO), an aggregation of several scoring functions into a penalty to measure deviation from a conservative baseline. This is viewed by reviewers as merging two formerly separate lines of work. Reviewers appreciated the evidence suggesting that weaker overseers can collectively constrain a stronger agent using the proposed framework, but raised questions regarding the generalizability and interpretation of results. In particular, they asked for additional ablation studies and comparison with competitive baselines. The authors provided extensive additional evidence in their rebuttal, resolving the reviewers' core concerns. Considering the broad positive opinions from all reviewers after the rebuttal, I recommend acceptance.